# Vision-Language Instruction Tuning:
# A Review and Analysis

**Chen Li** [†]                                                    *palchenli@tencent.com*

**Yixiao Ge** [†]                                                  *yixiaoge@tencent.com*

**Dian Li** [‡]                                                    *goodli@tencent.com*

**Ying Shan** [†]                                                  *yingsshan@tencent.com*

[†]*ARC Lab, Tencent PCG*
[‡]*Foundation Technology Center, Tencent PCG*

**Reviewed on OpenReview:** *https://openreview.net/forum?id=ul2tbUPtIQ*

## Abstract

Instruction tuning is a crucial supervised training phase in Large Language Models (LLMs), aiming to enhance the LLM's ability to generalize instruction execution and adapt to user preferences. With the increasing integration of multi-modal data into LLMs, there is growing interest in Vision-Language Instruction Tuning (VLIT), which presents more complex characteristics compared to pure text instruction tuning. In this paper, we systematically review the latest VLIT settings and corresponding datasets in multi-modal LLMs and provide insights into the intrinsic motivations behind their design. For the first time, we offer a detailed multi-perspective categorization for existing VLIT datasets and identify the characteristics that high-quality VLIT data should possess. By incorporating these characteristics as guiding principles into the existing VLIT data construction process, we conduct extensive experiments and verify their positive impact on the performance of tuned multi-modal LLMs. Furthermore, we discuss the current challenges and future research directions of VLIT, providing insights for the continuous development of this field. The code and dataset related to this paper have been open-sourced at `https://github.com/palchenli/VL-Instruction-Tuning`.

## 1 Introduction

Recently, there has been growing dissatisfaction with the limited capability of Large Language Models (LLMs) to process only discrete textual information (Brown et al., 2020; Chowdhery et al., 2022; Zhang et al., 2023c). As a result, researchers have been exploring techniques to enhance LLMs by enabling them to process additional modalities of information, primarily visual signals, allowing them to "see" beyond text (Huang et al., 2023a; Liu et al., 2023c; Zhu et al., 2023a). Integrating visual encoding modules into LLMs is currently the primary method for achieving this functionality, which improves the versatility of Multi-modal Large Language Models (MLLMs)[1] by facilitating the perception of visual data. Although the strategies employed by these MLLMs for visual signal conversion vary, their training paradigms follow a consistent pattern (i.e., pre-training and fine-tuning) (Liu et al., 2023c; Dai et al., 2023; Awadalla et al., 2023). Among these, fine-tuning with vision-language instructions plays a vital role throughout the training process.

Instruction tuning is a supervised training procedure that follows the pre-training stage in LLM training, typically encompassing a variety of tasks (Ouyang et al., 2022; Wei et al., 2021). This process offers a twofold benefit: enhancing the ability of LLMs to generalize and execute diverse task instructions while also

---

[1]In this paper, MLLMs specifically refer to LLMs capable of processing visual and text signals.

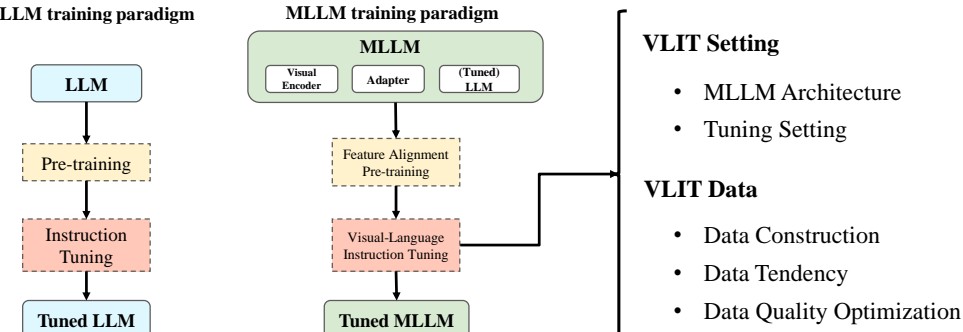

Figure 1: Different training paradigms of LLM and MLLM (left), and the VLIT-related key points collected and discussed in this paper (right).

bridging the gap between user preferences and model output. Vision-Language Instruction Tuning (VLIT) is a specialized adaptation of instruction tuning in a multi-modal scenario, tailored to the overall architecture design and training objectives of MLLMs. Compared to traditional instruction tuning, VLIT carries more responsibilities (Liu et al., 2023c; Dai et al., 2023; Zhu et al., 2023a). Numerous state-of-the-art MLLMs now demonstrate strong performance on zero-shot downstream visual tasks and boast exceptional user interaction capabilities, largely attributable to their efficient VLIT.

More specifically, as shown in Figure 1, VLIT facilitates the efficient tuning of MLLM, primarily due to two crucial components:

- **VLIT Setting** refers to the decision-making of whether each module in the MLLM architecture is tunable during the VLIT stage. Different MLLMs often integrate their unique frameworks and diverse understandings and requirements of VLIT to establish corresponding VLIT settings, guiding MLLMs to develop the desired capabilities.

- **VLIT Data**, as an element directly influencing MLLM performance, is of paramount importance in the VLIT process. On one hand, VLIT data must continuously guide MLLM in comprehending task instructions and user preferences. On the other hand, it also assumes the responsibility of directing MLLM to enhance cross-modal correlations further and understand more refined information based on instructions.

However, existing reviews of instruction tuning (Zhang et al., 2023d; Wang et al., 2023g) are typically limited to language-only LLMs. Discussions of instruction tuning in multi-modal scenarios remain scattered throughout various MLLM papers, lacking systematic summaries and comprehensive solutions (Zhou et al., 2023; Wei et al., 2023; Wang et al., 2023f; Xu et al., 2023a). Therefore, in this paper, we organize and summarize the latest MLLMs with VLIT stages and corresponding VLIT data, providing a reasonable classification and analysis of VLIT settings and data from different perspectives. First, we explore the underlying motivations hidden behind different VLIT settings. Simultaneously, *for the first time*, we analyze and outline the main characteristics that high-quality VLIT data should possess, formalizing them into a complete set of quantitative evaluation indicators for different perspectives. By thoroughly considering these principles at various stages (data collection, instruction-response generation, and quality control), we propose a pipeline capable of generating higher-quality VLIT data. To verify the effectiveness of these principles and the entire pipeline, we use common public annotation data to construct a VLIT dataset according to the aforementioned design and perform VLIT on three widely used MLLMs with different architectures (LLaVA (Liu et al., 2023c), BLIP-2 (Li et al., 2023h), and OpenFlamingo (Awadalla et al., 2023)) alongside two other classic VLIT datasets (LLaVA (Liu et al., 2023c) and MIMIC-IT (Li et al., 2023b)). Empirical results demonstrate that the VLIT data generated using the method proposed in this paper surpasses existing work. Moreover, the various evaluation indicators proposed in this paper exhibit a strong correlation with the output of tuned MLLMs in downstream tasks, and the quality control module implemented through these

**Pure Text Instruction**

Instruction: <instruction>

Input: <text>

Response: <output>

**Visual-Language Instruction**

<BOS> <context>

### Instruction: <instruction>

### Input: {<images>, <text>}

### Response: <output> <EOS>

Figure 2: Instruction data format examples of LLM and MLLM.

indicators can further enhance the performance of the tuned MLLMs. The code and VLIT data are both open source and will be regularly updated with new VLIT data.

In summary, the main contributions of this paper can be outlined as follows: 1) We present a systematic review of all relevant settings and datasets of VLIT in MLLMs. For the first time, we summarize the principles that should be considered when constructing VLIT data and highlight the challenges and directions that still need exploration. 2) We propose a comprehensive pipeline for constructing high-quality VLIT data and provide corresponding implementation details. 3) Based on the proposed construction pipeline and publicly available annotated data, we construct a VLIT dataset and conduct experiments comparing it with existing VLIT datasets on multiple MLLMs with different architectures. The results demonstrate the validity of the summarized principles and the effectiveness of the construction pipeline.

The remainder of this paper is organized as follows: Section 2 introduces the relevant background and preliminaries of VLIT. Section 3 organizes recent MLLMs and corresponding VLIT settings and datasets. Section 4 summarizes the VLIT data construction principles based on the aforementioned information and proposes a VLIT data construction pipeline and example implementation based on these principles. Section 5 primarily includes the design, implementation, and discussion of the verification experiment. Section 6 presents current challenges and future research directions.

## 2  Background and Preliminary

In this section, we provide a brief overview of the background and preliminaries of VLIT.

Recently, LLMs have developed the ability to perform diverse tasks based on different instruction descriptions, similar to humans. Instruction tuning is the primary technique employed to achieve this functionality. It introduces pre-defined instructions during the fine-tuning stage after pre-training, guiding LLMs to complete tasks while understanding the instructions. Compared with traditional supervised fine-tuning (Touvron et al., 2023a), which is limited to solving a single task, and prompt tuning (Lester et al., 2021; Liu et al., 2022), which guides LLMs only to strengthen a certain type of task, instruction tuning is not restricted to adapting to specific tasks but rather enhances its ability to solve any task. Many mature LLMs are equipped with instruction tuning, such as InstructGPT (Ouyang et al., 2022), ChatGPT (OpenAI, 2023a), and LLaMA (Touvron et al., 2023a). They follow pre-defined instructions effectively and support unseen new instructions, significantly improving zero-shot performance and greatly enhancing their generalization ability. In addition, some LLMs used as chatbots incorporate instruction tuning to make the output of LLMs more aligned with human preferences. They can adapt to real-world conversation scenarios while solving tasks and significantly improve their performance when directly interacting with users. This makes instruction tuning a critical training phase for enhancing the generalization and usability of LLMs.

There has been a surge in efforts to develop MLLMs due to the desire to go beyond accomplishing only discrete text-based tasks (Liu et al., 2023c; Zhu et al., 2023a; Awadalla et al., 2023; Dai et al., 2023). The introduction of visual signals has led to MLLMs being generally equipped with visual encoders and image-to-text adapters, and the training paradigm has been adjusted accordingly. Similar to the pre-training stage in LLM, MLLMs first prepare a large number of image-text pairs as training data to achieve feature alignment between modalities and convert visual signals into representations or tokens that LLMs can understand. Then, in the instruction tuning stage, there is an additional requirement for understanding visual information and

Table 1: The VLIT settings and VLIT data of MLLMs. The model architecture of MLLMs can be roughly divided into three parts (i.e., visual encoder, projector, and LLM), and the colors of the modules represent the two different settings that be trainable or frozen during the VLIT process. When adapting to multi-modal scenarios, the LLM needs to undergo two training stages: Pre-training (i.e., Feature Alignment (FA)) between modalities and Vision-Language Instruction Tuning (VLIT). Annotation Adaption (AA), Self-Instruct (SI), and Data Mixing (DM) represent the three paradigms for constructing VLIT data. Datasets marked with * indicate that they have undergone quality optimization. "-" represents that the module is not explicitly mentioned in the paper or does not exist. More detailed model and data statistics are provided in Table 8 and Table 9 in the appendix.

| MLLM | VLIT Setting | | | | VLIT Data | |
|---|---|---|---|---|---|---|
| | MLLM Architecture | | | Tuning | Dataset | DC |
| | Visual Encoder | Projector | LLM | Setting | | |
| Flamingo (Alayrac et al., 2022) | CLIP | Perceiver Resampler | LLM(XATTN) | FA | - | DM |
| KOSMOS-1 (Huang et al., 2023a) | CLIP$^2$ | Perceiver Resampler | MAGNETO | FA → VLIT | - | DM |
| LLaVA (Liu et al., 2023c) | CLIP | Linear | Vicuna | FA → VLIT | Instruct-150K | SI |
| Mini-GPT4 (Liu et al., 2023c) | Eva-CLIP | Q-Former+Linear | Vicuna | FA → VLIT | $cc\_sbu\_align^*$ | AA |
| mPLUG-Owl (Ye et al., 2023b) | CLIP | Linear | LLaMA | FA → VLIT | - | DM |
| VideoChat (Li et al., 2023j) | InternVideo | Q-Former+Linear | LLaMA | FA → VLIT | VideoChat | DM |
| VisionLLM (Wang et al., 2023e) | ResNet/Intern-H | BERTBase+D-DETR | Alpaca | FA → VLIT | - | DM |
| X-LLM (Chen et al., 2023c) | Eva-CLIP | Q-Former+Linear | ChatGLM | FA → VLIT | X-LLM | AA |
| DetGPT (Pi et al., 2023) | Eva-CLIP | Q-Former+Linear | Vicuna | FA → VLIT | DetGPT | SI |
| ChatBridge (Zhao et al., 2023e) | Eva-CLIP | Perceiver | Vicuna | FA → VLIT | MULTIS | AA |
| PandaGPT (Su et al., 2023) | ImageBind | Linear | Vicuna | VLIT | pandagpt_vid | DM |
| MVIT-TE/TD (Zhang et al., 2023f) | (PMC-)CLIP | Linear | (PMC-)LLaMA | FA → VLIT | PMC-VQA$^*$ | DM |
| UNIFIED-IO (Lu et al., 2023b) | VQ-GAN | - | T5 | FA → VLIT | - | DM |
| GPT4Tools (Yang et al., 2023) | Tools | - | OPT/LLaMA/.. | VLIT | GPT4Tools | SI |
| LLaVA-Med (Li et al., 2023e) | CLIP | Linear | Vicuna | FA → VLIT | - | AA |
| Ying-VLM (Li et al., 2023k) | Eva-CLIP | Q-Former+Linear | Ziya | FA → VLIT | $M^3IT^*$ | SI |
| Otter (Li et al., 2023b) | CLIP | Perceiver Resampler | LLaMA | VLIT | MIMIC-IT$^*$ | AA |
| Video-ChatGPT (Maaz et al., 2023) | CLIP | Linear | Vicuna | VLIT | Vid-ChatGPT$^*$ | SI |
| OFA$_{mnstruct}$ (Xu et al., 2022) | - | - | - | VLIT | MNSTRUCT$^*$ | AA |
| Video-LLaMA (Zhang et al., 2023a) | Eva-CLIP | (Vid)Q-Former+Linear | Vicuna | FA → VLIT | Video-LLaMA | DM |
| VALLEY (Luo et al., 2023b) | CLIP | Linear | Vicuna | FA → VLIT | Instruct-73k | SI |
| MM-GPT (Gong et al., 2023) | CLIP | Perceiver Resampler | LLaMA | VLIT | MM-GPT | AA |
| UNIFIED-IO2 (Lu et al., 2023a) | CLIP | Linear | Transformer | FA → VLIT | - | DM |
| InstructBLIP (Dai et al., 2023) | Eva-CLIP | Q-Former+Linear | FlanT5/Vicuna | VLIT | - | DM |
| LaVIN (Luo et al., 2023a) | CLIP | MMAdapter | LLaMA | VLIT | - | DM |
| MACAW-LLM (Lyu et al., 2023) | CLIP | MHSA | LLaMA | VLIT | MACAW-LLM | SI |
| LAMM (Yin et al., 2023) | CLIP | Linear | - | VLIT | LAMM | SI |
| OphGLM (Gao et al., 2023) | - | - | ChatGLM | VLIT | OphGLM | SI |
| LLaVAR (Zhang et al., 2023g) | CLIP | Linear | Vicuna | FA → VLIT | LLaVAR$^*$ | SI |
| Shikra (Chen et al., 2023e) | CLIP | Linear | Vicuna | FA → VLIT | Shikra-RD | SI |
| Polite Flamingo (Chen et al., 2023b) | CLIP | Perceiver Resampler | LLaMA/Guanaco | VLIT | PF-1M$^*$ | SI |
| mPLUG-DocOwl (Ye et al., 2023a) | CLIP | Linear | LLaMA | FA → VLIT | - | DM |
| GPT4RoI (Ye et al., 2023a) | CLIP | Linear | Vicuna | FA → VLIT | GPT4RoI | AA |
| SVIT (Zhao et al., 2023a) | CLIP | Linear | Vicuna | FA → VLIT | SVIT$^*$ | SI |
| KOSMOS-2 (Peng et al., 2023) | CLIP | Perceiver Resampler | MAGNETO | FA → VLIT | GRIT-20M | AA |
| BuboGPT (Zhao et al., 2023c) | Eva-CLIP | Q-Former+Linear | Vicuna | FA → VLIT | BuboGPT | DM |
| ChatSpot (Zhao et al., 2023b) | CLIP | Linear | Vicuna | FA → VLIT | MGVLID | DM |
| CoVLM (Li et al., 2023i) | CLIP+YOLOX | - | Pythia | FA → VLIT | - | DM |
| MMICL (Zhao et al., 2024) | Eva-CLIP | Q-Former+Linear | FLANT5 | FA → VLIT | MIC_5M | DM |
| ImageBind-LLM (Han et al., 2023) | ImageBind | Bind Network | LLaMA | FA → VLIT | - | DM |
| Lynx (Zeng et al., 2023b) | EVA | Resampler | Vicuna(Adapter) | FA → VLIT | Lynx | DM |
| LMEye(IPN) (Li et al., 2023m) | CLIP | Linear | OPT/Bloomz/.. | FA → VLIT | Multimodal | SI |
| (Gunjal et al., 2023) | - | - | - | - | M-HalDetect | SI |
| ASM (Wang et al., 2023d) | CLIP/RoIAlign | Linear | Husky | FA → VLIT | AS-1B$^*$ | DM |
| Cheetor (Li et al., 2023g) | CLIP | Q-Former+Linear | Vicuna/LLaMA2 | FA → VLIT | I4 | DM |
| BLIVA (Hu et al., 2023b) | Eva-CLIP | Q-Former+Linear | Vicuna | FA → VLIT | - | DM |
| (Hu et al., 2023a) | - | - | - | - | CIEM/CIT$^*$ | SI |
| StableLLAVA (Li et al., 2023l) | CLIP | Linear | Vicuna | FA → VLIT | StableLLAVA | AA |
| Qwen-VL (Bai et al., 2023) | CLIP | VL Adapter | Qwen | FA → VLIT | - | DM |
| VIGC (Wang et al., 2023a) | Eva-CLIP | Q-Former+Linear | Vicuna | VLIT | VIGC$^*$ | DM |
| PointLLM (Xu et al., 2023b) | Point-BERT$^3$ | Linear | Vicuna | FA → VLIT | PointLLM | SI |
| MLLM-DE (Zhao et al., 2023d) | Eva-CLIP | Q-Former+Linear | Vicuna | FA → VLIT | MLLM-DE$^*$ | SI |
| NExT-GPT (Wu et al., 2023) | ImageBind | Linear | Vicuna | FA → VLIT | T2M/MosIT | SI |
| PVIT (Chen et al., 2023a) | CLIP/RegionCLIP | Linear | Vicuna | FA → VLIT | PVIT | SI |
| TextBind (Li et al., 2023f) | Eva-CLIP | Q-Former+Linear | LLaMA2 | FA → VLIT | TextBind | SI |
| DreamLLM (Dong et al., 2023) | CLIP | Linear | Vicuna | FA → VLIT | DreamLLM | SI |
| AnyMAL (Moon et al., 2023) | CLIP/DINO | Linear | LLaMA2 | FA → VLIT | AnyMAL | AA |
| InternLM-XC (Zhang et al., 2023b) | Eva-CLIP | Perceive Sampler | InternLM | FA → VLIT | InternLM-XC | DM |
| SparklesChat (Huang et al., 2023b) | Eva-CLIP | Q-Former+Linear | Vicuna | FA → VLIT | SD-VG/VC | SI |
| LLaVA 1.5 (Liu et al., 2023b) | CLIP | Linear | Vicuna | FA → VLIT | - | DM |
| CogVLM (Wang et al., 2023c) | Eva-CLIP | SwiGLU | Vicuna+Expert | FA → VLIT | - | DM |
| Ferret (You et al., 2023) | CLIP+SAVSampler | Linear | Vicuna | FA → VLIT | Ferret | DM |
| Fuyu (Bavishi et al., 2023) | - | Linear | FuYu Transformer | FA → VLIT | - | - |
| MiniGPT-v2 (Chen et al., 2023d) | Eva-CLIP | Linear | LLaMA2 | FA → VLIT | MiniGPT-v2 | DM |
| (Du et al., 2023) | - | - | - | - | ComVint | SI |
| GLaMM (Rasheed et al., 2023) | CLIP+SAM | Linear | Vicuna | FA → VLIT | GranD | SI |
| SEED-LLaMA (Ge et al., 2023) | CLIP | Causal Q-Former+CB | Vicuna/LLaMA2 | FA → VLIT | - | DM |
| OtterHD (Li et al., 2023a) | - | Linear | FuYu Transformer | VLIT | - | DM |
| mPLUG-Owl2 (Li et al., 2023a) | Eva-CLIP | Q-Former+Linear | LLaMA2 | FA → VLIT | - | DM |
| (Wang et al., 2023b) | - | - | - | - | LVIS-INST4V | SI |
| VideoPoet (Kondratyuk et al., 2024) | MAGVIT-v2 | - | -$^4$ | FA → VLIT | - | DM |

instructions beyond the original responsibilities. As shown in Figure 2, the main difference between current mainstream vision-language instructions and pure text instructions is that images are included as one of the input information sources.

Currently, as a recently emerged and rapidly evolving research area, some recent works have started to summarize existing VLIT studies. Among them, due to the consistent training objectives of VLIT with the pre-training stage, the differences in its settings are primarily reflected in the update strategies of various module parameters within the MLLM framework. Comparatively, more work concentrates on the relationship between VLIT data quality and the performance of tuned MLLMs. However, research on pure text instructions is limited to methodological discussions, lacking both quantitative and qualitative experimental validation, and does not provide guidance for constructing and filtering instructional tuning data (Zhang et al., 2023d; Wang et al., 2023g; Lou et al., 2023). Furthermore, the analysis of VLIT data is only sporadically present in some MLLM literature, without offering a comprehensive and reliable mature experience (Li et al., 2023k;b; Du et al., 2023).

Here, we give a formal definition of VLIT. Given a vision-language instruction sample ($Instruction$, $Input$, $Response$), where $Instruction$ is the instruction sentence, $Input = \{<images>, <text>\}$ is the input images and context text, $Response$ is the response of the given instruction and input. The MLLM predicts the answer of the given instruction and input, i.e., $Predicted = \text{MLLM}(Instruction, Input; \theta)$, where $\text{MLLM}(\cdot)$ denotes the MLLM, and $\theta$ is the parameters of MLLM. Further, in line with the autoregressive training setting in the pre-training stage, the instruction tuning stage still needs to correctly predict the next token in $Response$ as the training objective.

Then, the loss function of the instruction tuning stage can be written as

$$\mathcal{L}(\theta) = -\sum_{i=1}^{N} \log \ p(Response_{[i+1]}|Instruction, Input, Response_{[0:i]}; \theta). \tag{1}$$

## 3 A Review of Vision-Language Instruction Tuning

In this section, we thoroughly sort out all VLIT-related work, analyzing the MLLM architecture, instruction tuning paradigm, and the corresponding tuning data. For relevant statistics, please refer to Table 1, and corresponding detailed information can be found in Table 8 and Table 9 in the appendix.

### 3.1 Setting for Vision-Language Instruction Tuning

The VLIT stage is an important step for adjusting MLLM, especially for further linking visual signals with the LLM. In this section, we first briefly introduce the basic architecture and existing characteristics of MLLM. Then, we summarize the settings for different components in the architecture during the VLIT process.

#### 3.1.1 MLLM Architecture

As illustrated in Figure 2 and Table 1, the current MLLM structure primarily comprises three parts.

- **Visual encoder**, serving as the visual signal entry point for MLLM, is mainly responsible for encoding the image. As shown in Table 1, CLIP (Radford et al., 2021) and its variant version: EVA-CLIP (Fang et al., 2023) are selected as the visual encoder for the vast majority of MLLMs due to its powerful visual representation and text alignment capabilities. In addition, classic or novel visual encoders such as ResNet (He et al., 2015) and ImageBind (Han et al., 2023) are also selected by some MLLMs due to model compatibility. Overall, as the entry point for MLLM to receive visual signals, the visual encoder usually only needs to be able to convert visual information such as images and videos into vector representations to meet the requirements.

- **LLM** processes various input information and generates response content. Recently, it is the gradual maturity of LLMs that has attracted a lot of attempts to understand multi-modal information (especially visual) using LLMs. Although both are seq2seq models, these LLMs can be divided into three categories based on their architectures. Among them, Encoder-only architecture LLMs are better at analyzing and classifying text content, such as BERT (Devlin et al., 2019) and RoBERTa (Liu et al., 2019) adopt this framework. LLMs that use Decoder-only architecture are classic autoregressive

models that consider previous inputs as context and output subsequent content. As shown in Table 1, LLaMA (Touvron et al., 2023a), vicuna (a version of LLaMA that uses more text data for tuning) (Zheng et al., 2023), OPT (Zhang et al., 2022), and LLaMA2 (Touvron et al., 2023b), which use this architecture, are the choices for most MLLMs. In addition, LLMs based on the Encoder-Decoder framework, such as T5FLAN (Chung et al., 2022) and ChatGLM (Zeng et al., 2023a), use new technologies and architecture adjustments to optimize performance, especially for tasks with complex mapping relationships between input and output sequences. Therefore, they are also chosen by some MLLMs.

- **Projector** is tasked with connecting the above two relatively mature modules by querying and projecting the representation of the input visual signal, enabling the information output by the visual encoder to be processed by LLM. Considering the explicit function of the projector and its unrestricted implementation methods, there are many ways to implement it. As shown in Table 1, many methods use the most direct linear projection layer (Liu et al., 2023c; Su et al., 2023) or projection matrix (Wu et al., 2023) or pixel-level linear projection (Bavishi et al., 2023; Li et al., 2023a) to map visual semantics and have achieved good results. Some methods (Zhu et al., 2023a; Pi et al., 2023) directly employ Q-Former (stacked transformers) trained on three different visual-text tasks in BLIP2 (Li et al., 2023h) and concatenate them to the visual encoder to selectively extract visual information in the form of queries. Similarly, the Perceiver Resampler (Huang et al., 2023a; Li et al., 2023b) in Flamingo (Alayrac et al., 2022) and some adapters designed based on the transformer structure (Wang et al., 2023e; Lyu et al., 2023; Han et al., 2023; Wang et al., 2023c) are also used to selectively map input visual representations.

## 3.2 Tuning Setting

To equip the aforementioned MLLM with the capability to understand and process multi-modal data, it undergoes approximately two training processes, similar to LLM: pre-training and fine-tuning, as shown in Table 1. In contrast to LLM pre-training, which is based on massive text, the "pre-training" stage of MLLM typically relies on image-text pairs to achieve the initial projection from visual semantics to text semantics, which can be understood as cross-modal feature alignment. In the fine-tuning stage (i.e., VLIT stage), MLLM assumes more responsibilities than LLM. On one hand, it needs to continue guiding LLM to understand additional visual data and instructions. On the other hand, it must further direct LLM to associate instructions with different granularity visual data to provide support for successfully executing corresponding instructions. As demonstrated in Table 8, all MLLMs essentially follow the above training paradigm. Some methods have also made fine adjustments to the training process to enhance the performance of VLIT. For instance, some work further decomposes the tuning steps by subjectively defining the before and after relationships of tasks, aiming to progressively endow MLLM with new capabilities (Wang et al., 2023d; Bai et al., 2023; Zhang et al., 2023b; Chen et al., 2023d).

Additionally, as shown in Table 1, it is worth noting that not all module parameters will be updated within the entire VLIT framework. Some MLLMs (Liu et al., 2023c; Wang et al., 2023e; Li et al., 2023e; Luo et al., 2023b; Zhao et al., 2023c;b; Bai et al., 2023; Xu et al., 2023b; Chen et al., 2023a; Li et al., 2023f; Dong et al., 2023; Chen et al., 2023d), represented by LLaVA, typically make their adapters and LLM tunable during VLIT to achieve the main goal of VLIT. That involves continuing to tune the adapter to further adapt to the information conversion between modalities and opening LLM tuning to improve its understanding of instructions, associating instructions with fine-grained visual information, and ultimately completing the ability to execute instructions. Simultaneously, some methods employ LoRA (Hu et al., 2022) instead of full tuning to control the scale of LLM updates, preventing the impact of VLIT on the original capabilities of LLM (Su et al., 2023; Ye et al., 2023b; Yang et al., 2023; Li et al., 2023k; Gong et al., 2023; Yin et al., 2023; Han et al., 2023; Zhao et al., 2023d; Tu et al., 2023; Wu et al., 2023; Moon et al., 2023). Moreover, some other MLLMs directly freeze the LLM parameters and only update the adapter parameters (Chen et al., 2023c; Pi et al., 2023; Zhang et al., 2023f; Maaz et al., 2023; Zhang et al., 2023a; Lyu et al., 2023; Liu et al., 2023a; Zeng et al., 2023b; Huang et al., 2023b), as the adapter possesses the ability to understand instructions (Dai et al., 2023; Li et al., 2023g; Hu et al., 2023b; Wang et al., 2023a) or directly insert visual information into the LLM structure through the adapter (Zhao et al., 2023e; Zhang et al., 2023b; Wang et al., 2023c; You et al.,

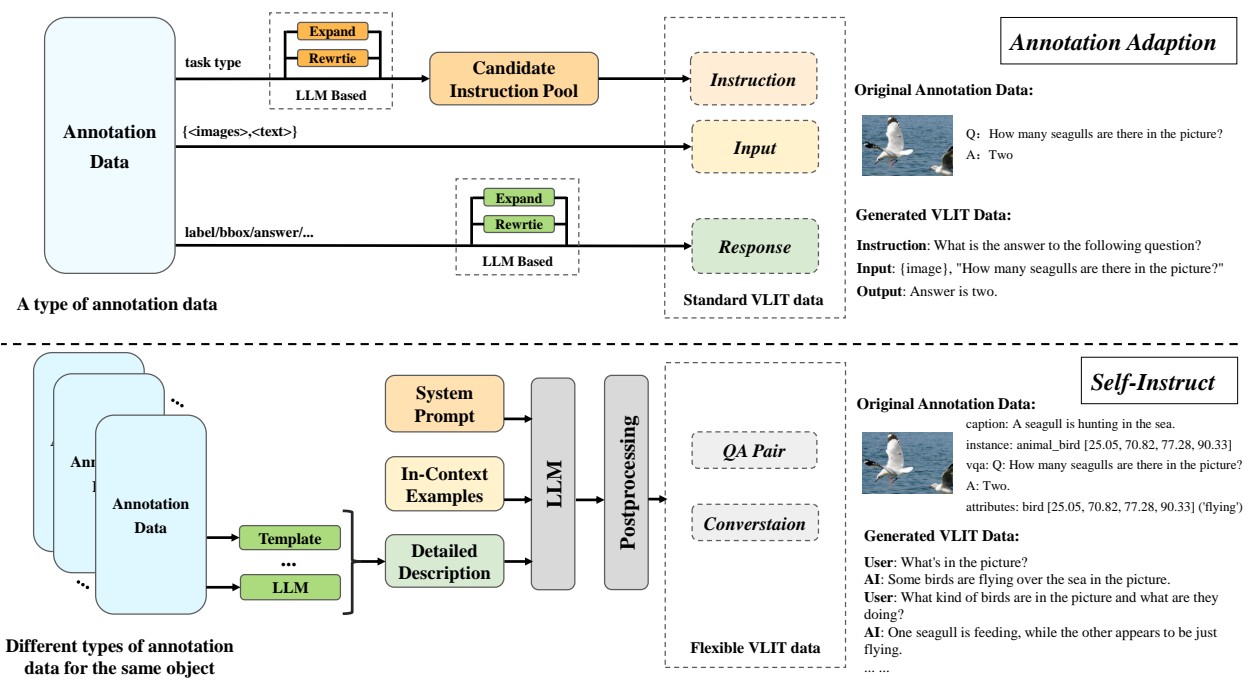

Figure 3: Two different construction paradigms for VLIT data: Annotation Adaption (AA) and Self-Instruct (SI). Specifically, AA usually only considers one type of annotation data as input, while SI considers multiple types of annotation data as input. AA independently considers attributes such as instruction and reply diversity to output standard VLIT data, while SI considers all different types of annotated information jointly to generate interactive VLIT data, such as QA pairs or multi-turn conversations. The toy examples on the right are the outputs of two different construction paradigms.

2023). Lastly, there are some MLLMs that aim to achieve VLIT from the visual encoder Li et al. (2023j) and even open the tuning of all modules (Peng et al., 2023), which can also yield positive application results.

## 3.3   Data for Vision-Language Instruction Tuning

As the key to guiding MLLMs to enhance their ability to understand instructions and generalize instruction execution, VLIT data has consistently been the focus of all MLLM-related work. In this section, we categorize existing work from two independent perspectives: data construction strategies and instruction tendencies. The overall classification of the tuning data can be observed in Figure 6.

### 3.3.1   Construction Strategy

VLIT data generally comes from a variety of publicly annotated visual task data (such as object detection, VQA, etc.). When constructing VLIT data, it is usually designed based on the type of input annotated data, the way the data is organized, and the style of the output data. Specifically, as illustrated in Figure 3, it can be divided into two categories: Annotation Adaption and Self-Instruct. In addition to the above two, mixing existing data and newly constructed data (i.e., Data Mixing) of different styles is also a common way to expand VLIT data.

**Annotation Adaption**   In recent years, the rapid advancement of visual models has led to the creation of large-scale, diverse, and high-quality annotated data (Lin et al., 2014; Patterson & Hays, 2016; Veit et al., 2016; Antol et al., 2015). This data is suitable for a wide range of downstream tasks and can be easily adapted for use as instructional data. As shown in Figure 3, many studies specifically tailor the annotated

data to adhere to the standard format found in instructional data. The *Instruction* component, which serves as a formal statement explaining the task's nature, is typically derived from manually written candidate instruction pools (Zhu et al., 2023a; Wang et al., 2023e; Chen et al., 2023c; Zhang et al., 2023f; Li et al., 2023k; Xu et al., 2022; Dai et al., 2023; Ye et al., 2023a; Zhang et al., 2023e; Peng et al., 2023; Zhao et al., 2023b; Zeng et al., 2023b; Moon et al., 2023; You et al., 2023). Some researchers use manual instructions as seeds to guide LLMs in rewriting (Zhu et al., 2023a) or expanding (Wang et al., 2023e; Zhao et al., 2023e; Gong et al., 2023; Zeng et al., 2023b) the instruction pool and enhancing instructional diversity. The *Input* consists of images and optional context, generally obtained from the original annotation data, and forms an integral part of the instructional data. The *Response* corresponds to the result found in the annotation data and represents the output portion of the instructional data. If the original annotation data meets the user's needs, it is usually transferred without any alterations. However, it is crucial to recognize that for specific downstream tasks, such as classification, judgment, and brief captions, which typically generate outputs of only words or short sentences, the original answer may need to be expanded based on input information using LLMs as necessary to prevent potential overfitting and other related issues (Zhu et al., 2023a; Zhang et al., 2023f; Zeng et al., 2023b). Notably, some studies consider such concise responses as a form of instruction and thus choose to incorporate suitable limiting language in the instruction to address this concern.

**Self-Instruct**   The scale, diversity, and creativity of instructions are often constrained by existing tasks and corresponding annotation data, hindering the universality of MLLMs that are tuned using this data. The standard instruction format faces challenges in adapting to evolving real-world application scenarios and addressing user needs. Therefore, to obtain additional instructional data that is more relevant to real-world situations, Self-Instruct is initiating the integration of a broader variety of annotation data (Wang et al., 2022). This approach will prompt LLMs to generate a more diverse and content-rich range of instruction-following data. Specifically, such methods leverage the powerful understanding and generation capabilities of LLMs. As illustrated in Figure 3, they organize the annotation information (e.g., caption, bbox, label, etc.) into a detailed description of the image based on the template, LLM, etc., and then guide the language-only LLM to generate QA pairs or directly into multi-round conversations. Among them, to guide LLM correctly, these methods will manually customize detailed system prompts and some examples and further improve the quality of generated content through in-contextual learning (Liu et al., 2023c; Pi et al., 2023; Li et al., 2023e;b; Yin et al., 2023; Liu et al., 2023a; Zhang et al., 2023g; Li et al., 2023l; Zhao et al., 2023d; Wu et al., 2023; Chen et al., 2023a; Li et al., 2023f; You et al., 2023) (Zhao et al. believes that this will limit diversity). Self-Instruct eliminates strict restrictions on the data, allowing for increased diversity and complexity in both instructions and responses.

**Data Mixing**   Of course, to expand the scale of VLIT data while ensuring that VLIT data has good diversity and other properties, many methods use the above two methods at the same time and construct large-scale VLIT data. Recently, some MLLMs have even merged VLIT data from multiple sources with different origins to further improve the performance of MLLMs, as shown in Figure 6.

### 3.3.2   Instruction Tendency

Due to the excellent ability of MLLM to process multi-modal information, some works have begun to explore its capacity to understand and execute more granular and domain-specific data. As shown in Figure 6, we have conducted statistics based on the design of specific objects, forms, or domains in different VLIT datasets.

Among them, the main focus of the VLIT data related to specific objects is on different granularity or forms of visual information. For example, for data with different granularity, some works explore the performance of MLLM on data types such as single region, multiple regions, and image-text interleaving. Some works, represented by (Zhang et al., 2023g), focus on images with rich text information and enhance their performance on text-related tasks by constructing VLIT data. There is also a large number of video-related data, which introduces the modeling of temporal information into the VLIT process.

In addition, there are corresponding VLIT data and MLLMs in fields such as medicine (Li et al., 2023e; Zhang et al., 2023f; Gao et al., 2023), documents (Ye et al., 2023a), and point clouds (Xu et al., 2023b).

These works combine their domain-specific data and instruction characteristics to propose VLIT data that is more suitable for their scenarios.

### 3.3.3 Data Quality Optimization

In general, the VLIT data constructed by the methods outlined in Sec. 3.3.1 may contain a significant number of defects, particularly inherent issues (such as hallucination and expression ambiguity) introduced by LLM during the construction process. These issues tend to be concentrated in the generated VLIT data. As shown in Table 1, to mitigate these problems as much as possible, enhance data quality, and further improve the performance of the tuned MLLM, numerous researchers have optimized the construction of VLIT data based on their understanding.

Firstly, to ensure the accuracy of individual VLIT data samples, some methods employ manual annotation (Zhu et al., 2023a; Li et al., 2023k; Xu et al., 2022; Wang et al., 2023d), while others utilize classification based on the causes of correctness issues (Zhao et al., 2023d) (such as hallucination (Wang et al., 2023a), recognition errors (Zhang et al., 2023e), external knowledge (Zhang et al., 2023f), etc.). Secondly, from a broader perspective, some methods emphasize the diversity of VLIT data from various angles. For instance, numerous methods enrich the types of tasks by designing additional tasks (Dai et al., 2023; Li et al., 2023k). Some methods introduce pure text instructions (Ye et al., 2023b; Luo et al., 2023a; Lyu et al., 2023; Peng et al., 2023; Zeng et al., 2023b; Bai et al., 2023) and rewrite or expand existing instructions (as mentioned in Sec. 3.3.1) to enhance the diversity of instructions within a single task. Other methods also control the length of the output text to consider the diversity of responses (Gong et al., 2023; Liu et al., 2023a). Thirdly, many researchers have started to recognize the importance of instruction complexity. Consequently, some methods attempt to increase the complexity of VLIT data by introducing multi-round dialogues (Lyu et al., 2023), adding negative samples (Yang et al., 2023; Liu et al., 2023a; Zhao et al., 2023c; You et al., 2023), and incorporating more complex task formats (Xu et al., 2023b). Lastly, some studies acknowledge the significance of balance in VLIT and introduce methods for balancing distributions within VLIT data (Zhao et al., 2023a; Zeng et al., 2023b).

However, there are two main problems in the construction of the current VLIT dataset.

- Most existing VLIT data construction methods are based on intuition and some scattered unverified practical experience, and lack reasonable, reliable, and systematic construction principles to guide the construction of VLIT data (especially in visual data).

- The current instruction data filtering strategy is still very rough and cannot quantitatively and qualitatively evaluate a single data sample and the overall data distribution. As a result, the existing VLIT data is generally very large and noisy, which greatly affects the downstream performance and tuning efficiency of MLLMs.

## 4 How to Construct High-Quality VLIT Data?

In this section, we begin by meticulously summarizing the characteristics of high-quality VLIT data. Subsequently, we integrate these characteristics as guiding principles into the existing construction process, thereby forming a pipeline that generates superior-quality VLIT data. Simultaneously, we also provide a sample implementation of this pipeline and construct a VLIT dataset.

### 4.1 Characteristics and Principles

Combining the discussion in Sec. 3.3.3, we describe in detail the characteristics that high-quality VLIT data should possess and organize them into several principles as follows:

First of all, as the data input to MLLM to train MLLM, it must be usable, so that it will not have a negative impact on MLLM. Then the principles of usable can be summarized as:

**Principle 1.** *Readability: The text and images in VLIT data need to support MLLM obtaining sufficient information, which means that the text and images need to have semantic readability.*

**Principle 2.** *Correctness: The visual information and textual content in each VLIT data (i.e., instruction-response pair) must be perfectly matched, ensuring that there are no inaccuracies in the descriptions of visual information or external knowledge within the generated content.*

Diversity has long been a focal point in natural language generation and a crucial aspect of evaluating content quality (Tevet & Berant, 2020; Zhang et al., 2020). In particular, diversity plays a critical role in the tuning of LLM instructions, as it determines the LLM's ability to generalize instructions (Zhang et al., 2023d). Specifically, in the field of VLIT, diversity exhibits more distinct characteristics, which can be summarized into the following three aspects:

**Principle 3.** *Task Diversity: VLIT data should encompass an adequate variety of tasks, encompassing both textual and visual tasks.*

**Principle 4.** *Instruction Diversity: In VLIT data, it is essential for each task to comprise a sufficient variety of distinct instruction sentences.*

**Principle 5.** *Response Diversity: In VLIT data, the response text for each task needs to maintain sufficient distinguishability.*

In particular, in VLIT, the diversity of visual information will also directly affect the performance of MLLM in executing instructions in different scenarios. Therefore, the diversity of visual information should also be taken into consideration.

**Principle 6.** *Visual Semantic/Object Diversity: The visual information (i.e., visual semantics and objects) between different VLIT data needs to have sufficient differences to ensure that VLIT has sufficient diversity as a whole.*

LLM with text instructions is the task's complexity, which has resulted in the development of the chain of thought tuning. In a multi-modal scenario, high-quality VLIT data with appropriate complexity can further guide MLLMs to jointly comprehend visual and textual semantics at varying granularities. Consequently, when constructing VLIT, two complexity-related principles must be taken into account:

**Principle 7.** *Instruction Complexity: In VLIT data, it is essential to ensure that some instructions necessitate a sufficiently complex logic for completion.*

**Principle 8.** *Object/Granularity Complexity: In VLIT data, the granularities of the instruction's object operation must exhibit a sufficient level of complexity.*

At the same time, since complex visual information often increases the difficulty of understanding and executing instructions, it is also necessary to consider the complexity of visual information in VLIT data to ensure that it can assist MLLMs in achieving efficient VLIT.

**Principle 9.** *Visual Semantic Complexity: The visual information contained in VLIT data needs to include more complex scenes so that VLIT can guide MLLMs to process instructions at different semantic granularities.*

Maintaining data balance is crucial for both traditional machine learning and LLMs. Research has shown that uneven data settings may lead to forgetting phenomena in large models, seriously hindering their overall performance (Wei et al., 2023; Zhai et al., 2023). Therefore, the following principle related to balance is equally important.

**Principle 10.** *In VLIT data, the distribution of tasks needs to be uniform and there is no obvious long tail.*

## 4.2 Principle Guided VLIT Data Construction

Based on the aforementioned principles, we incorporate corresponding considerations into each step of the classic VLIT data generation pipeline and added filtering measures that cover all principles in the final quality control module. In the following sections, we provide a comprehensive description and an example implementation of the VLIT data generation pipeline guided by these principles.

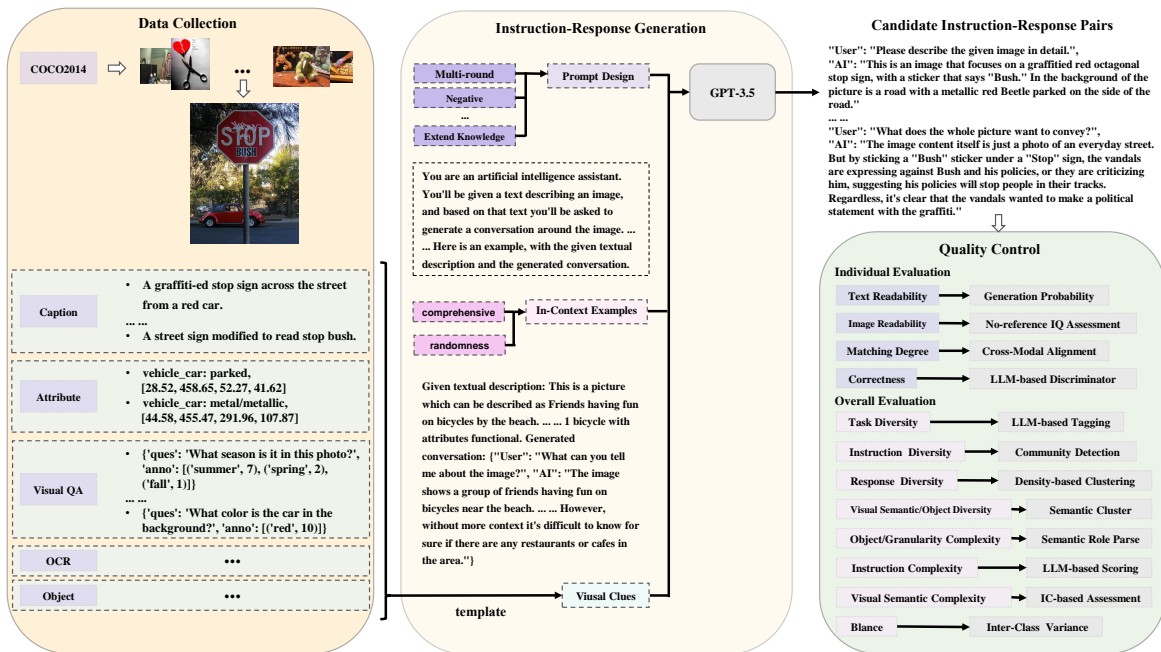

Figure 4: The complete pipeline for VLIT data construction. Data collection (refer to Sec. 4.2.1) is the stage of collecting annotated data; Instruction-Response Generation (refer to Sec. 4.2.2) uses GPT3.5 to generate multi-turn conversations based on preset system prompts, manually edited in-context examples, and collected visual clues; Quality control (refer to Sec. 4.2.3) filters the generated data based on the principles proposed in Sec. 4.1.

### 4.2.1 Data Collection

Multi-modal annotation data used to generate VLIT data typically originates from various visual tasks, such as recognition, segmentation, and classification. These tasks often serve as the source of supervision for training traditional machine learning models, providing large-scale, high-quality data, which makes them highly practical.

When collecting data, it is essential to thoroughly consider factors such as the type, source, scale, and quality of the annotation data. This can ensure that the collected information generates VLIT data that meets the requirements of correctness, diversity, and complexity. Specifically, annotation data that has been manually reviewed and widely used has higher credibility and will not cause subsequent errors due to annotation inaccuracies. Simultaneously, different types of annotation data can usually be extended to define various tasks, significantly enhancing the diversity of the final VLIT data. When multi-modal data from the same source contains different types of annotations, it can even generate more complex contexts and new task types on top of the original simple visual tasks, ensuring complexity.

In this paper, we provide a simple implementation to quickly construct VLIT data, as shown in Figure 4. Specifically, in data collection, we first select COCO 2014 (Lin et al., 2014) as the image source, and $\{caption, object, attribute, OCR, visualQA\}$ as the selected sources of annotation data (Antol et al., 2015; Patterson & Hays, 2016; Veit et al., 2016).

### 4.2.2 Instruction-Response Generation

Currently, using LLMs (such as ChatGPT, GPT4, and LLaMA) to assist in generating VLIT data has become mature. In contrast, annotation adaption is significantly less flexible, so we refer to self-instruct to generate VLIT data. Specifically, when providing an image and corresponding annotation data of various types, it is necessary to design appropriate templates for each type of annotation data to ensure that the inserted

annotation data conforms to standard language representation. By combining all annotation data sources, a detailed and accurate description of the image can be obtained, which is an understandable visual clue for language-only LLMs. To ensure the diversity and complexity of the generated content, it is necessary to write sufficiently detailed prompts to guide the LLMs, and manually write a sufficient number of example instruction-response pairs. In the system prompt design, we follow the principles proposed in Sec. 3.3.3 and guide the generator to generate more diversified and complex instruction-response pairs by flexibly using system prompts with different additional constraints (e.g., extended knowledge, negative samples, multi-round dialogues, etc.). At the same time, the context examples we design are sufficiently sufficient, meet the characteristics of comprehensiveness and randomness, correspond to the boundaries involved, and provide appropriate examples of all possible scenarios so that the generator can correctly understand the meaning of the prompts. In addition, to ensure sufficient diversity during generation, we also redundantly set the key points of the system prompts and context examples, that is, repeat certain key points multiple times (such as not generating uncertain content, avoiding fuzzy answers, etc.).

In implementation, we employ GPT-3.5 (OpenAI, 2023a) as the generator for LLM, and formulate corresponding system cues and in-context examples[5]. Specifically, based on the annotated data and different system prompts, we introduce different granularities of tasks, add negative samples, generate multi-turn complex dialogues, and extend external knowledge in the generation process to enhance the diversity and complexity of the generated VLIT data. In addition, to further enhance diversity, we set the temperature value of GPT-3.5 to 0.8 and the number of generations to 5.

### 4.2.3  Quality Control

Owing to the potential errors in LLM itself and the uncontrollable nature of the generated data distribution, it is crucial to conduct quality control on the VLIT data directly output by LLM. Recently, numerous studies have started to address this issue and have filtered the data based on established experience (Li et al., 2023b;k). In this paper, we design appropriate data filtering strategies based on the principles proposed in Sec. 4.1, and carry out local and global quality control on the generated VLIT data.

**Usability.**  Regarding the readability mentioned in Principle 1, we need to conduct readability analysis separately for the text and images in VLIT data. For text readability, we use the overall generation probability of the text calculated by a trained language model (e.g., Word2Vec (Mikolov et al., 2013)) to achieve quantitative evaluation.

$$Readability_{text} = \frac{1}{N} \sum_{i=1}^{N} \log(P_i). \tag{2}$$

For image readability, we focus on the overall clarity of the image, so we refer to a related mature visual task (i.e., no-reference image quality assessment, which aims to assess the perceptual quality of images by human subjective perception) and corresponding method (i.e., MANIQA (Yang et al., 2022)) for quantitative evaluation. In accordance with the two aspects mentioned in Principle 2, it is vital to assess the degree of alignment between visual and textual information in the given candidate VLIT data, as well as the overall accuracy. In this paper, we opt for the classic pre-training model, CLIP (Radford et al., 2021), to implement the matching function. Specifically, we calculate the representations of visual and textual information separately and evaluate the degree of alignment through similarity. For accuracy evaluation, we introduce MLLM (i.e., LLaVA 1.5 (Liu et al., 2023b)) and manually written guiding examples to conduct multiple accuracy assessments on the given VLIT data, taking the highest frequency as the final accuracy score.

**Diversity.**  To prevent the influence of the inherent correlation of original tasks themselves (e.g., detection and classification) when evaluating task diversity, we attempt to establish new task categories for the generated instructions. In practice, we employ a zero-shot labeler TagGPT (Li et al., 2023d) to assign category labels to the instructions based on their content. Among them, TagGPT is a labeler that can be cold-started. It generates labels to samples by introducing the semantic understanding capability of LLM, retains high-quality labels based on indicators such as label coverage and redundancy, and finally assigns labels to samples. When

---

[5]See the Appendix for all system prompts used during the generation process

evaluating instruction diversity, since there is still some semantic similarity in the instruction text for tasks with the same task type, it is not possible to directly use semantic calculations to evaluate instruction diversity. Although some methods try to avoid this problem by counting predicate verbs, some noun information also affects the calculation of diversity. In this paper, we use a graph-based method to calculate instruction diversity, namely, converting instruction texts with the same task label into an element graph based on co-occurrence relationships after removing stop words and using the overlapping community partitioning method (i.e., Ego-Splitting (Epasto et al., 2017)) to count the number of clusters to evaluate instruction diversity. Ego-Splitting is a generative model with overlapping clusters in which it can reconstruct most of the clusters exactly by using the connected component algorithm as a non-overlapping partition algorithm both in the ego-net clustering phase and in the persona graph clustering phase. When evaluating response diversity, we directly utilize an efficient unsupervised text representation method (i.e., SimCSE (Gao et al., 2021)) to represent response texts and employ a density-based clustering algorithm (i.e., DBScan (Schubert et al., 2017)) to mine semantic clusters as indicator. SimCSE generates unsupervised positive samples by randomly sampling dropout masks to improve contrastive learning performance. That is, during the training phase, the same sample is input into the same encoder twice to obtain two different representation vectors $z, z'$, $z'$ will be used as a positive sample, then the training target of the model is:

$$l_i = -\log \frac{e^{sim(h_i^{z_i}, h_i^{z_i'})/\tau}}{\sum_{j=1}^{N} e^{sim(h_i^{z_i}, h_j^{z_j'})/\tau}}. \tag{3}$$

DBScan is a clustering method that treats clusters as dividing high-density areas by low-density areas in the data space, and divides sufficiently high-density areas into clusters. It cycles through labeling candidate samples to corresponding clusters and deleting them, and finally achieves clustering after all samples are labeled. To ensure that images can also provide rich and diverse semantic information as mentioned in Principle 6, we first refer to the evaluation method of response diversity. We employ a mature visual encoder (i.e., CLIP) to obtain image representations, and evaluate visual semantic diversity by clustering (e.g., DBScan) these visual representations and counting the number of clusters. In addition, to evaluate the number of different object categories in the images, we utilize an open-set object detection algorithm (e.g., YOLO-World (Cheng et al., 2024)) to count the number of objects and evaluate the object diversity of the images.

**Complexity.** To evaluate complexity, we first use classical semantic dependency analysis to count the number of elements in the instructions (Màrquez et al., 2008), and then utilize LLaMA-2 as the discriminative model to provide a quantitative difficulty score under the guidance of the given sample. Specifically, semantic dependency analysis refers to the task of extracting important semantic elements in a text and assigning corresponding roles. In this paper, we use mature tools[6] for semantic role extraction as a data source for counting the number of instruction elements. When evaluating the complexity of instructions, similar to existing work, we input given instruction-reply pairs into LLM (i.e., LLaMA-2) and rely on manually scored examples to give all candidate pairs a complexity score of 1-10. For the evaluation of visual semantic complexity mentioned in Principle 9, we also refer to a classic visual task (i.e., automatic image complexity assessment, which is a task that simulates human subjective evaluation of image complexity.) and use a corresponding mature solution for visual complexity evaluation (Feng et al., 2022).

**Balance.** When assessing balance, we can utilize the obtained task labels to calculate the variance of sample sizes across different tasks, serving as a balance indicator. Simultaneously, by randomly and uniformly sampling from various task labels, we can acquire VLIT data with category balance.

## 5 Experiments

In this section, we will thoroughly examine the correlation between the evaluation indicators proposed in Sec 4.2.3 and the performance of tuned MLLMs. Specifically, we select two common VLIT data sets and

---

[6]https://hanlp.hankcs.com/demos/sdp.html

the generated VLIT data in Sec 4.1, amounting to a total of three types of VLIT data, which are used for instruction tuning of MLLMs with varying architectures. Subsequently, by jointly analyzing the evaluation indicators of the VLIT data and their performance in different tuned MLLMs, we can ascertain the validity and applicability of these evaluation indicators.

## 5.1 Implementation Details

We list the relevant settings of these experiments in detail as follows.

### 5.1.1 Multi-Modal Large Language Model

To fairly verify the influence of instruction data on tuning results, we choose three MLLMs with different frameworks, respectively:

**LLaVA** (Liu et al., 2023c), is a representative MLLM consisting of a visual encoder, an LLM, and a trainable linear projection layer. It shows good session performance after two-stage training (i.e., pre-trained feature alignment and end-to-end instruction tuning). In this paper, we use LLaVA library[7] for implementation. We select LLaVA-7B as MLLM, in which the visual encoder is CLIP (ViT-L-14) (Radford et al., 2021) and the LLM is Vicuna-7B[8]. And we refer to the original paper setting and directly use the open-source checkpoint[9] as the pre-trained MLLM.

**BLIP-2** (Li et al., 2023h), a new structure Q-former is proposed to connect visual information with textual information. After Q-former has been trained on three pre-defined image-text tasks (i.e., matching, contrastive, and generation), visual information can be extracted by trainable queries and combined with the fully connected layer to realize the conversion of visual features to textual features. In this paper, we use its official library, i.e., LAVIS[10] and utilize checkpoints pre-trained on the COCO-Caption as the initial MLLM. Its visual encoder is CLIP and its language model is OPT-2.7B[11] (Zhang et al., 2022).

**OpenFlamingo** (Awadalla et al., 2023), is an open-source version[12] of Flamingo (Alayrac et al., 2022). It includes a visual information conversion module composed of a visual encoder and perceiver resampler, which can convert images into a few visual tokens and put them into LLM. In this paper, we use its official version: OpenFlamingo-3B, in which the visual encoder is CLIP (ViT-L-14) and the LLM is OPT-1.3[13] (Zhang et al., 2022). And we utilize the checkpoint[14] of OpenFlamingo-3B, which is pre-trained on MMC4 (Zhu et al., 2023b) and LAION (Schuhmann et al., 2022), as the initial MLLM.

These MLLMs are all trained utilizing 8 Telsa V100 (32G) GPUs with the Python environment and other detailed settings (e.g., hyperparameters) of the three models can be found in Section A.3 in the Appendix.

### 5.1.2 Vision-Language Instruction Tuning Data

To validate the correlation between the quality of VLIT data and the proposed characteristics, as well as the utility of the suggested VLIT data construction pipeline, we utilize the example VLIT data constructed in Sec.4.1 and select LLaVA (LLaVA-Instruct-150K)(Liu et al., 2023c) and MIMIC-IT (Li et al., 2023b) as comparative data. In the quality evaluation process, to ensure fairness, we use the smallest dataset size as the scale for all the test VLIT data and randomly sample VLIT datasets larger than this scale. In addition to incorporating insights on high-quality VLIT data during construction, we also rely on the designed quality control strategy to filter the sample data constructed in this paper, further verifying the accuracy and validity of the conclusions presented in this paper[15].

---

[7]https://github.com/haotian-liu/LLaVA.

[8]https://huggingface.co/lmsys/vicuna-7b-v1.1.

[9]https://huggingface.co/liuhaotian/llava-pretrain-vicuna-7b-v1.3.

[10]https://github.com/salesforce/LAVIS.

[11]https://huggingface.co/facebook/opt-2.7b.

[12]https://github.com/mlfoundations/open_flamingo.

[13]https://huggingface.co/facebook/opt-1.3b.

[14]https://huggingface.co/openflamingo/OpenFlamingo-3B-vitl-mpt1b.

[15]See Sec. B in the Appendix for more generated instruction-response examples

Table 2: Performance of different VLIT data in three MLLMs. *w/o* means without.

| MLLM | VLIT Data | Overall | SU | II | IL | IA | IC | SR | IIR | VR | TR |
|---|---|---|---|---|---|---|---|---|---|---|---|
| LLaVA | LLAVA | 28.0 | 25.3 | 28.3 | 33.9 | 24.3 | 25.8 | 27.6 | 10.0 | 10.3 | 18.2 |
| | MIMIC-IT | 26.3 | 24.4 | 25.4 | 30.7 | 24.3 | 23.1 | 25.7 | 10.0 | 14.8 | 21.5 |
| | Ours | 28.3 | 25.0 | 27.0 | 35.0 | 24.3 | 25.8 | **26.3** | 10.0 | 17.2 | 45.5 |
| | Ours(QC) | **28.7** | **26.3** | **29.1** | **36.1** | **26.5** | 26.4 | **26.3** | 20.0 | 18.3 | **47.1** |
| | *-w/o usability* | 28.4 | 25.7 | 28.7 | 35.8 | **26.5** | 26.9 | **26.3** | 20.0 | 17.8 | 46.3 |
| | *-w/o diversity* | 28.6 | 25.0 | 28.7 | 35.8 | **26.5** | 26.4 | 25.7 | 20.0 | 19.1 | **47.1** |
| | *-w/o complexity* | 28.5 | 25.7 | 27.0 | **36.1** | 25.2 | 26.4 | 25.7 | 20.0 | 19.1 | **47.1** |
| | *-w/o balance* | 28.3 | **26.3** | 28.7 | 33.9 | 22.7 | 26.4 | 24.6 | 20.0 | **20.6** | 40.3 |
| BLIP-2 | LLaVA | 27.3 | 25.7 | 25.3 | 33.9 | 22.7 | 22.4 | 22.4 | 20.0 | 10.3 | 8.6 |
| | MIMIC-IT | 26.5 | 24.1 | 24.4 | 31.2 | 20.1 | 21.7 | 21.8 | 20.0 | 13.8 | 9.1 |
| | Ours | 27.5 | 26.3 | 26.5 | 33.9 | 25.2 | 23.6 | 23.7 | 20.0 | 13.8 | 9.1 |
| | Ours(QC) | **28.4** | **27.7** | 27.0 | **35.0** | 25.2 | 25.8 | **25.7** | 20.0 | **19.4** | **17.5** |
| | *-w/o usability* | 27.8 | 26.3 | 27.0 | 34.4 | 25.2 | 25.8 | 24.6 | 20.0 | 17.2 | 15.9 |
| | *-w/o diversity* | 28.2 | 26.3 | 26.5 | **35.0** | 25.2 | 26.4 | **25.7** | 20.0 | 17.8 | **17.5** |
| | *-w/o complexity* | 28.0 | 25.7 | 27.0 | 34.4 | **26.5** | 26.4 | **25.7** | 20.0 | 18.3 | **17.5** |
| | *-w/o balance* | 27.1 | **27.7** | **28.7** | 33.9 | 24.3 | 23.1 | 23.7 | 20.0 | **19.4** | 11.7 |
| OpenFlamingo | LLaVA | 25.5 | 25.7 | 27.0 | 30.7 | 22.7 | 23.6 | 25.2 | 10.0 | 13.8 | 21.5 |
| | MIMIC-IT | 25.8 | 23.9 | 24.4 | 33.9 | 22.7 | 24.0 | 24.6 | 20.0 | 10.8 | 20.4 |
| | Ours | 28.1 | 28.2 | 26.0 | 33.9 | 21.4 | 25.5 | 22.4 | 20.0 | 17.2 | 27.3 |
| | Ours(QC) | **29.1** | 30.5 | **29.6** | **37.9** | **26.5** | 27.5 | **24.6** | 20.0 | **20.6** | **30.7** |
| | *-w/o usability* | 28.6 | 29.4 | 29.6 | 37.1 | **26.5** | 26.4 | **24.6** | 20.0 | 19.1 | 28.9 |
| | *-w/o diversity* | 28.9 | 29.4 | 29.1 | **37.9** | **26.5** | 28.1 | 23.7 | 20.0 | 19.8 | **30.7** |
| | *-w/o complexity* | 28.6 | 28.7 | 28.7 | 37.1 | **26.5** | 26.4 | 23.7 | 20.0 | 19.8 | 27.3 |
| | *-w/o balance* | 28.4 | **31.7** | **29.6** | 36.1 | 24.3 | 25.5 | 22.4 | 20.0 | **20.6** | 15.9 |

### 5.1.3 Metrics

To more comprehensively evaluate the understanding and generation capabilities of MLLMs, we employ SEED-Bench (Li et al., 2023c) as the evaluation metric for MLLMs after instruction tuning. SEED-Bench comprises a large number of multiple-choice questions, and we have selected nine tasks related to vision-language tasks for evaluation. Specifically, as shown in Table 3, "SU" represents Scene Understanding, "II" denotes Instance Identity, "IL" signifies Instance Location, "IA" stands for Instance Attribute, "IC" indicates Instance Counting, "SR" refers to Spatial Relation, "IIR" corresponds to Instance Interaction, "VR" represents Visual Reasoning, and "TR" denotes Text Recognition.

### 5.1.4 Comparison of VLIT Data

The quality assessment of three different VLIT data and the performance of different MLLMs tuned with them can be seen in Table 2 and Table 3 respectively.

Firstly, by observing the results in Table 2, we can discern that the example VLIT data constructed in this paper significantly outperforms the other two in the majority of metrics. Simultaneously, the VLIT data, after being filtered by the quality control strategy, exhibit better performance, demonstrating that the characteristics of high-quality VLIT data summarized in this paper are accurate, and the quality control scheme designed in the text is reasonable. Second, we perform ablation experiments separately by ignoring different filtering metrics (i.e., usability(refer to Principle 1,2), diversity(refer to Principle 3,4,5), complexity(refer to Principle 7,8), and balance(refer to Principle 10)) in quality control filtering. The experimental results of three MLLMs show that filtering based on the proposed metrics can indeed improve the overall performance of MLLM. When carefully observing the changes in each sub-task indicator in Table 2, it can be found that when different types of filter indicators are removed, the corresponding related task evaluation indicators will

Table 3: Quality assessment results of different VLIT data. Among them, the Text Readability (TR), the Matching Degree (MD) and Correctness (C) of a single sample is the average of all samples in VLIT data; the Task (T) diversity index is the number of tasks, the Instruction (I) diversity index is the number of clusters in the element graph, and the Response (R) diversity index is the log value of the number of spatial semantic clusters; the Object/Granularity (O/G) complexity index is the number of objects, the Instruction (I) complexity is the average score given by LLM; the balance index is the variance of the number of samples in different tasks. ↑ represents the higher the better, while ↓ represents the lower the better. ICC is the inter-class correlation coefficient, and Single Measurement (SM) and Average Measurement (AM) are two kinds of measurements.

| Dataset | Quality Evaluation | | | | | | | | ICC | |
| | Single | | | Overall | | | | | | |
| | Usability | Correctness | | Diversity | | | Complexity | | | | |
| | TR ↑ | MD ↑ | C ↑ | T ↑ | I ↑ | R ↑ | O/G ↑ | I ↑ | Balance ↓ | SM | AM |
|---|---|---|---|---|---|---|---|---|---|---|---|
| LLaVA | 0.84 | 30.7 | 91.1 | 9 | 14.5 | 2.2 | 1.6 | 6.5 | 14.6 | 0.8083 | 0.9267 |
| MIMIC-IT | 0.83 | 30.9 | 93.4 | 9 | 13.3 | 2.3 | 1.3 | 4.4 | 10.5 | 0.7268 | 0.8887 |
| Ours | 0.84 | 33.2 | 94.6 | 10 | 20.5 | 2.3 | 1.9 | 6.8 | 16.8 | 0.3036 | 0.5667 |
| Ours(QC) | 0.84 | 34.1 | 94.8 | 10 | 22.6 | 2.5 | 2.0 | **7.1** | **12.3** | 0.4281 | 0.6920 |
| *-w/o usability* | 0.84 | 33.1 | 94.6 | 10 | 22.5 | 2.5 | 2.0 | **7.1** | 12.7 | 0.4084 | 0.6744 |
| *-w/o diversity* | 0.84 | 34.1 | **94.9** | 10 | 20.5 | 2.3 | 1.9 | 7.0 | 12.4 | 0.4374 | 0.6999 |
| *-w/o complexity* | 0.84 | 34.1 | 94.8 | 10 | 21.8 | 2.4 | 1.9 | 6.9 | 12.4 | 0.3950 | 0.6620 |
| *-w/o balance* | 0.84 | **34.3** | **94.9** | 10 | **22.8** | 2.5 | **2.3** | **7.1** | 16.7 | 0.3212 | 0.5867 |

Table 4: Performance of VLIT data under different visual information qualities in three MLLMs.

| Setting | IR | VD | | VSC | Overall | | |
| | | SD | OD | | LLaVA | BLIP-2 | OpenFalmingo |
|---|---|---|---|---|---|---|---|
| Original | 0.5343 | 44 | 5.5 | 48.9 | 27.6 | 27.0 | 27.8 |
| Readability ↓ | 0.2245 | 44 | 5.5 | 48.9 | 24.3 | 23.9 | 24.7 |
| Diversity ↓ | 0.5339 | 27 | 3.3 | 48.6 | 26.8 | 26.3 | 27.2 |
| Complexity ↓ | 0.5295 | 43 | 5.3 | 42.2 | 27.4 | 26.7 | 27.7 |

change. The more obvious ones are "diversity & instruction recognition" and "complexity & scene recognition". In addition, due to the removal of the "balance" filter indicator, some sub-tasks (e.g., visual reasoning and sense understanding) account for a larger proportion of the VLIT data, which indirectly leads to a significant improvement in the performance of the corresponding tasks. At the same time, the performance of the tasks corresponding to a smaller proportion of data decreases. To ensure the robustness and credibility of the experimental results, we evaluate the consistency (i.e., the inter-class correlation coefficient (Koo & Li, 2016)) of the performance of the same VLIT data in different MLLM frameworks, and the results are shown in Table 3. It can be observed that the performance of the same dataset in different MLLMs is essentially or highly consistent. This further ensures the reliability of the conclusions we derive.

In particular, since the three VLIT datasets we selected are all based on high-quality data sets mainly including COCO, the evaluation results of their visual data (i.e., image readability, diversity, and complexity) are basically the same, making them unsuitable for direct comparative analysis. Therefore, we choose to discuss it independently. To investigate the impact of images with different qualities on VLIT data, we filtered the dataset constructed in the paper and manually constructed VLIT data containing images of different qualities, as shown in Table 4. Specifically, we used image distortion (corresponding to setting 1), adjusted the number of clusters and objects (corresponding to setting 2), and re-splitter based on a predetermined visual semantic complexity threshold (corresponding to setting 3) to form several equally sized subsets of VLIT

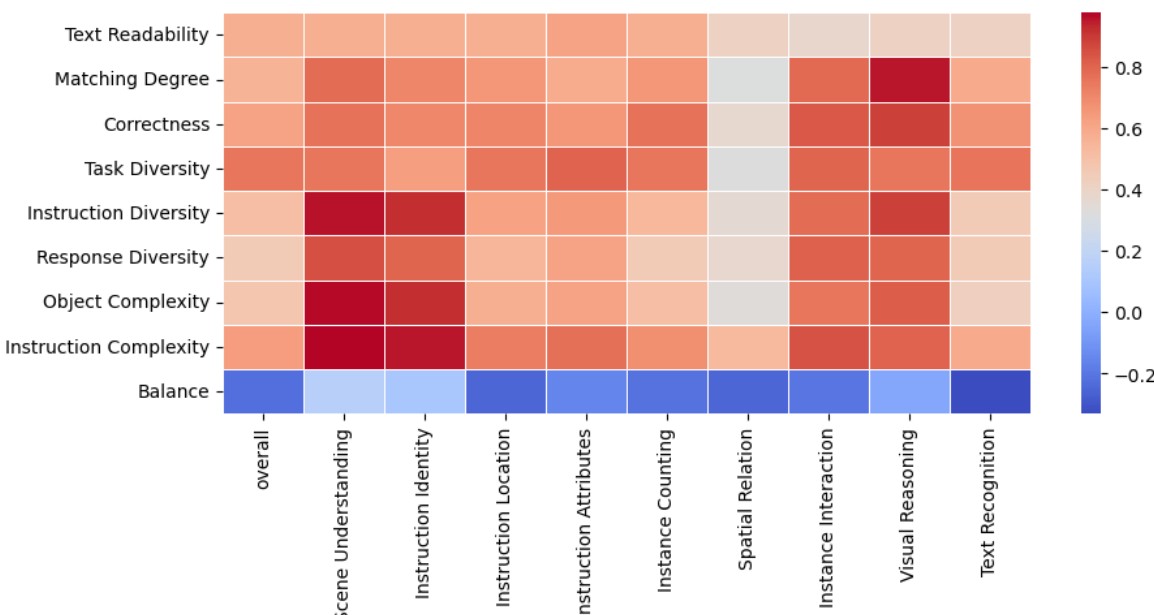

Figure 5: Correlation analysis between the quality control indicators of VLIT data and the performance of tuned MLLM in different downstream tasks.

data for training three baseline MLLMs. The experimental results based on VLIT data of different qualities are shown in Table 4. It can be observed that VLIT data of higher quality have improved VLIT performance to varying degrees, demonstrating the crucial role of high-quality visual data in VLIT performance. Among them, the most important is the readability of the image, followed by the diversity of the image, and the complexity of the image has relatively little impact on VLIT.

Simultaneously, as shown in Table 3, we compile statistics on the performance of the mentioned VLIT data under the evaluation indicators proposed in Sec.4.1. To further verify the rationality of the proposed evaluation indicators, we conduct a correlation analysis between each quality control indicator and the performance of MLLM on different tasks after utilizing its tuning. Since it has been demonstrated above that all VLIT data exhibit good consistency across three different MLLMs, we use the performance mean of VLIT data on three MLLMs for analysis, and the results are displayed in Figure 5. Upon observation, we can discern that the results of most quality evaluation indicators are directly proportional to the performance of the tuned MLLM on downstream tasks (balance is inversely proportional because lower indicators indicate better quality). Except for the spatial relationship indicator, there is only a certain correlation between object/granularity complexity and instruction complexity. This may be attributed to the fact that spatial positional relationships typically only have a certain correlation with the number of elements appearing in the instruction and the related instructions.

# 6 Challenges and Future Directions

Despite the substantial amount of data and research available concerning vision-language instruction tuning, the field remains in its nascent stages. Consequently, there is still much to explore and numerous challenges to overcome. We highlight several pertinent directions for future investigation.

**Mature Multi-Modal Large Language Model.** Recently, with the disclosure of GPT-4V (OpenAI, 2023b), a large number of MLLMs have matured visual understanding capabilities, and they no longer need to use text to describe visual signals for language-only LLMs to generate VLIT. However, we still need to note that hallucinations, information loss, and other phenomena are more serious in MLLM than in LLM.

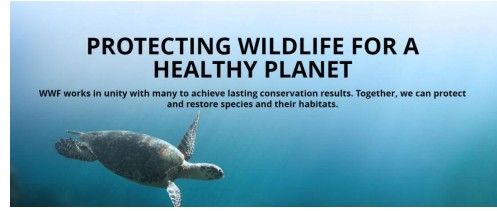

**GPT4V**: The image displays a serene underwater scene with a focus on marine conservation. The backdrop consists of a gradient of blue hues, representing the ocean's depth, with a lighter blue at the top transitioning to darker shades toward the bottom, suggesting the increasing depth of the sea.
Centrally positioned in the image is a sea turtle, angled such that it appears to be swimming upward toward the left side of the frame. The turtle is captured in clear detail, showing its patterned shell, flippers, and head, all characteristic of a sea turtle's anatomy.
Overlaid on the image, in the upper portion, is bold white text that reads "PROTECTING WILDLIFE FOR A HEALTHY PLANET." This is a clear statement emphasizing the theme of wildlife protection being crucial for the health of our planet.
Below the headline, in smaller white text, is an additional message that states, "WWF works in unity with many to achieve lasting conservation results. Together, we can protect and restore species and their habitats." This further emphasizes the purpose of the World Wide Fund for Nature (indicated by the WWF acronym), highlighting their collaborative efforts in conservation and restoration of wildlife and habitats.

**Error**:
1. the color of the text,
2. the position of the turtle,
3. the direction of the turtle's movement.

Table 5: An example of GPT4V incorrect description of the given image. The location of the error has been marked with an underline in the original text.

| | |
|---|---|
| 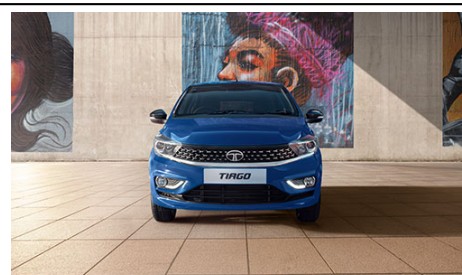 | **Instruction**: Please indicate what is on the license plate of the car in the picture. |
| | **Mini-GPT4**: The license plate on the car says "GO". ✗
**LLaVA 1.5**: The license plate on the car says "TIGO." ✗
**Qwen-VL**: The license plate of the car is not visible in the image, so we cannot determine what it says. ✗ |
| 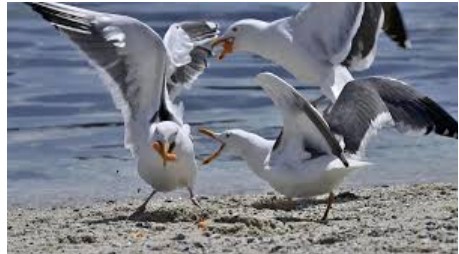 | **Instruction**: Please answer the question: "Are there more birds flying in the picture or birds standing on the ground?". |
| | **Mini-GPT4**: There are three seagulls standing on the sandy beach, with one of them holding its beak open and the other two looking at it. The water in the background is calm and blue. There are no other birds in the picture. ✗
**LLaVA 1.5**: There are more birds standing on the ground in the picture than flying. ✓
**Qwen-VL**: There are more birds flying in the sky than standing on the ground in the picture. ✗ |

Table 6: Difficult examples (fine-grained content understanding and complex reasoning) for existing commonly used MLLMs.

For example, as shown in the examples in Table 5, although GPT4V can describe the image very well and in detail, it still has some errors. Therefore, how to reasonably use only MLLM to guide the generation of VLIT data and achieve quality control is a direction worth studying.

**Difficult Sample.** Existing MLLMs have demonstrated strong instruction execution and generalization abilities in some simple tasks. However, as shown in Table 6, they generally have shortcomings when dealing with some special difficult samples, such as fine-grained content understanding, complex reasoning, and multi-modal content editing (i.e., image/video editing). The chain-of-thought (Wei et al., 2022) and in-context learning (Dong et al., 2022) can alleviate the occurrence of the above problems to some extent, but the cost of obtaining high-quality annotated data is too high, and the model computation cost it brings is also unacceptable. Therefore, how to efficiently solve such difficult samples is a problem worthy of attention.

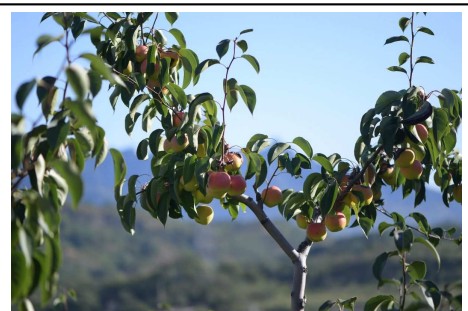

| | |
|---|---|
| | **Instruction**: Please describe this image in detail. |
| | **Mini-GPT4**: This image shows a tree with several ripe peaches hanging from its branches. The tree is in a field with mountains in the background. The peaches are a bright red color and appear to be ripe and ready to be picked. The leaves on the tree are a deep green color and there are some small flowers on the branches. The sky is a bright blue and there are a few clouds in the distance. The overall mood of the image is peaceful and serene. |

Table 7: An example of hallucination phenomena that MLLMs experience when describing an image (mistaking apples for peaches and hallucinating flowers and clouds).

**Selective Forgetting.** As pointed out by Zhai et al.., fine-tuning LLMs can easily cause them to forget their original capabilities (e.g., the classification ability of the original CLIP) or instructions, which is catastrophic. In fact, this phenomenon is prevalent in LLM, and abundant research has demonstrated that supervised fine-tuning in downstream tasks frequently causes overfitting of the LLM in the corresponding downstream tasks, leading to the loss of some or all of its original capabilities. Currently, existing studies generally address this issue by regulating the learning rate and minimizing the overall loss, achieving certain outcomes. The training data in instructional tuning serves as the learning object of the model. It is equipped with appropriate settings, such as the repetition of old samples and the balancing of task ratios, to mitigate the forgetting phenomenon to a certain extent. Nonetheless, further research is necessary to tackle this matter.

**Hallucination and Bias.** Similar to LLM (OpenAI, 2023a; Touvron et al., 2023a), the MLLM model is prone to hallucinations and biases. As shown in Table 7, Hallucinations occur when the MLLM generates non-existent content, as it unconsciously imitates data in its training corpus. In the multi-modal environment, this issue has resulted in the creation of non-existent visual targets, which can significantly mislead users. The issue of bias arises when content generated caters towards specific groups or delivers prejudiced outcomes, compromising user experience, impartiality, and equality, among other aspects. This is typically the result of partial training data distribution. To address the problem, this paper suggests employing additional, more detailed image-text fully aligned samples, numerous negative samples, as well as data quality modules. However, obtaining corresponding data on a large scale can often be difficult due to the absence of detailed annotations.

**Limited Emergence.** As a significant purpose of instruction tuning, instruction generalization is an advantage that LLM (and MLLM) should possess, stemming from the emergence phenomenon. In the multimodal scenario, emergence shows more characters, such as supporting visual objects that have not appeared. However, recently, a large number of MLLMs have not shown good performance in either of the above two scenarios. Therefore, deeper consideration is needed on how to design MLLMs and corresponding instruction tuning data.

## 7 Conclusion

In this paper, we have compiled recent research on vision-language instruction tuning, presenting succinct summaries and analyses, along with categorization, details, and shortcomings of existing literature. To construct high-quality vision-language instruction data, we suggested several construction principles and a complete set of construction processes based on the existing research summaries. Extensive experimentation has demonstrated that MLLMs, when tuned on the instruction data constructed via our proposed method, yield better overall performance. To conclude, we lay out the remaining issues and potential future research directions.

Table 8: MLLMs with VLIT stage. "Training Paradigm" refers to the training phase included in the MLLM, where FA is short for feature alignment (MLLMs lacking this phase either use the pre-trained ckpt directly or do not have this phase). "Pure Text" refers to whether the VLIT data contains pure text instructions. "Data Source" refers to whether the data used for instruction tuning contains instruction data from other existing VLIT data. "Self Construction" refers to whether this MLLM is tuned by VLIT data constructed autonomously.

| Name | Training Paradigm | Pure Text | Data Source | Self Construction | URL Model | Data |
|---|---|---|---|---|---|---|
| LLaVA (Liu et al., 2023c) | FA → VLIT | ✗ | Pure | ✓ | url | url |
| Mini-GPT4 (Liu et al., 2023c) | FA → VLIT | ✗ | Pure | ✓ | url | url |
| mPLUG-Owl (Ye et al., 2023b) | FA → VLIT | ✓ | Hybrid | ✗ | url | - |
| VideoChat (Li et al., 2023j) | FA → VLIT | ✗ | Hybrid | ✓ | url | url |
| VisionLLM (Wang et al., 2023e) | FA(Object Detection) → VLIT | ✗ | Hybrid | ✗ | url | - |
| X-LLM (Chen et al., 2023c) | FA(X2L) → VLIT | ✗ | Hybrid | ✗ | url | url |
| DetGPT (Pi et al., 2023) | FA → VLIT | ✗ | Pure | ✓ | url | url |
| ChatBridge (Zhao et al., 2023e) | FA → VLIT | ✗ | Hybrid | ✓ | url | url |
| PandaGPT (Su et al., 2023) | VLIT | ✗ | Hybrid | ✗ | url | url |
| MedVInT-TE/TD (Zhang et al., 2023f) | VLIT | ✗ | Pure | ✓ | url | url |
| GPT4Tools (Yang et al., 2023) | VLIT | ✗ | Pure | ✓ | url | url |
| LLaVA-Med (Li et al., 2023e) | FA → VLIT | ✗ | Pure | ✓ | url | - |
| Ying-VLM (Li et al., 2023k) | FA → VLIT | ✗ | Hybrid | ✓ | - | url |
| Otter (Li et al., 2023b) | VLIT | ✗ | Hybrid | ✓ | url | url |
| Video-ChatGPT (Maaz et al., 2023) | VLIT | ✗ | Pure | ✓ | url | url |
| OFA$_{multiinstruct}$ (Xu et al., 2022) | VLIT | ✗ | Pure | ✓ | url | url |
| Video-LLaMA (Zhang et al., 2023a) | FA → VLIT | ✗ | Hybrid | ✗ | url | url |
| VALLEY (Luo et al., 2023b) | FA → VLIT | ✗ | Pure | ✓ | url | url |
| MultiModal-GPT (Gong et al., 2023) | VLIT | ✓ | Hybrid | ✗ | url | url |
| InstructBLIP (Dai et al., 2023) | VLIT | ✗ | Hybrid | ✓ | url | - |
| LaVIN (Luo et al., 2023a) | VLIT | ✓ | Hybrid | ✗ | url | - |
| MACAW-LLM (Lyu et al., 2023) | VLIT | ✓ | Pure | ✓ | url | url |
| LAMM (Yin et al., 2023) | VLIT | ✗ | Pure | ✓ | url | url |
| OphGLM (Gao et al., 2023) | VLIT | ✓ | Pure | ✓ | url | - |
| LLaVAR (Zhang et al., 2023g) | FA → VLIT | ✗ | Pure | ✓ | url | url |
| Shikra (Chen et al., 2023e) | FA → VLIT | ✗ | Hybrid | ✓ | url | url |
| Clever Flamingo (Chen et al., 2023b) | FA → VLIT | ✗ | Pure | ✓ | url | url |
| mPLUG-DocOwl (Ye et al., 2023a) | FA → VLIT | ✓ | Hybrid | ✓ | url | - |
| GPT4RoI (Ye et al., 2023a) | FA(Region Adaption) → VLIT | ✗ | Pure | ✓ | url | url |
| SVIT (Zhao et al., 2023a) | FA → VLIT | ✗ | Pure | ✓ | url | url |
| KOSMOS-2 (Peng et al., 2023) | FA → VLIT | ✓ | Hybrid | ✓ | url | url |
| BuboGPT (Zhao et al., 2023c) | FA → VLIT | ✗ | Hybrid | ✗ | url | url |
| ChatSpot (Zhao et al., 2023b) | FA → VLIT | ✗ | Pure | ✓ | - | - |
| ImageBind-LLM (Han et al., 2023) | FA(Parameter-efficient Tuning) → VLIT | ✓ | Hybrid | ✗ | url | url |
| Lynx (Zeng et al., 2023b) | FA → VLIT | ✓ | Hybrid | ✓ | url | url |
| LMEye(IPN) (Li et al., 2023m) | FA(Image-Text Matching) → VLIT | ✗ | Hybrid | ✓ | url | url |
| ASM (Wang et al., 2023d) | FA → VLIT(Multi-Task Tuning) | ✗ | Hybrid | ✓ | url | url |
| Cheetor (Li et al., 2023g) | FA → VLIT | ✗ | Pure | ✓ | url | url |
| BLIVA (Hu et al., 2023b) | FA → VLIT | ✗ | Hybrid | ✗ | url | - |
| StableLLAVA (Li et al., 2023l) | FA → VLIT | ✗ | Pure | ✓ | url | url |
| Qwen-VL (Bai et al., 2023) | FA → VLIT(Multi-Task Tuning) | ✓ | Pure | ✓ | url | - |
| VIGC (Wang et al., 2023a) | VLIT | ✗ | Hybrid | ✓ | url | url |
| PointLLM (Xu et al., 2023b) | FA → VLIT | ✗ | Pure | ✓ | url | url |
| MLLM-DataEngine (Zhao et al., 2023d) | VLIT | ✗ | Hybrid | ✓ | url | url |
| MM (Tu et al., 2023) | FA → VLIT | ✗ | Hybrid | ✗ | url | - |
| NExT-GPT (Wu et al., 2023) | FA → VLIT(Modality-Switching) | ✓ | Pure | ✓ | url | url |
| PVIT (Chen et al., 2023a) | FA → VLIT | ✗ | Pure | ✓ | url | url |
| LLaVA(MS+IR+DM) (Lu et al., 2023c) | FA → VLIT | ✓ | Hybrid | ✗ | - | - |
| TextBind (Li et al., 2023f) | FA → VLIT | ✗ | Pure | ✓ | url | url |
| DreamLLM (Dong et al., 2023) | FA(Interleaved Generative) → VLIT | ✗ | Hybrid | ✓ | url | - |
| AnyMAL (Moon et al., 2023) | FA → VLIT | ✓ | Pure | ✓ | - | - |
| InternLM-XComposer (Zhang et al., 2023b) | FA → VLIT(Multi-Task Tuning) | ✓ | Hybrid | ✓ | url | - |
| SparklesChat (Huang et al., 2023b) | FA → VLIT | ✗ | Hybrid | ✓ | url | url |
| LLaVA 1.5 (Liu et al., 2023b) | FA → VLIT | ✓ | Hybrid | ✓ | - | - |
| CogVLM (Wang et al., 2023c) | FA → VLIT | ✗ | Hybrid | ✗ | url | - |
| Ferret (You et al., 2023) | FA → VLIT | ✗ | Hybrid | ✓ | url | - |
| MiniGPT-v2 (Chen et al., 2023d) | FA → VLIT(Multi-Task Tuning) | ✗ | Hybrid | ✓ | url | url |
| GLaMM (Rasheed et al., 2023) | FA → VLIT | ✗ | Pure | ✓ | url | url |
| SEED-LLaMA (Ge et al., 2023) | FA → VLIT | ✓ | Hybrid | ✗ | url | - |
| OtterHD (Li et al., 2023a) | FA → VLIT | ✗ | Hybrid | ✗ | url | - |
| mPLUG-Owl2 (Li et al., 2023a) | FA → VLIT | ✗ | Hybrid | ✗ | url | - |

# A  Appendix

## A.1  Details of VLIT Setting and Data

For more details, please see Table 8, Table 9, and Figure 6.

## A.2  Vision-Language Tasks

We summarise the visual tasks covered in all current MLLMs and datasets. Since the names used to describe tasks are inconsistent across different tasks, we unified similar tasks as much as possible. All statistical results are shown in Table 10.

Table 9: VLIT datasets. "G" (Global), "R" (Regional), and "I" (Interleaved) represent the different compositions of the image, and "I-R" is the abbreviation of Instruction-Response. "Type" refers to the strategy used for data construction, where "AA" indicates annotation adaption, and "SI" indicates Self-Instruct. "-" indicates the absence or absence of clear information.

| Name | Model | Inst | Img | | | #Imgs | Vid | #Vids | #I-R |
|---|---|---|---|---|---|---|---|---|---|
| | | | G | R | I | | | | |
| LLaVA-Instruct-150K (Liu et al., 2023c) | LLaVA | SI | ✓ | ✓ | ✗ | 80K | ✗ | - | 158K |
| cc_sbu_align (Zhu et al., 2023a) | Mini-GPT4 | AA | ✓ | ✗ | ✗ | 3.5K | ✗ | - | 3.5K |
| VideoChat (Li et al., 2023j) | VideoChat | SI | ✓ | ✓ | ✗ | 7K | ✓ | 11K | 18K |
| X-LLM (Chen et al., 2023c) | X-LLM | AA | ✓ | ✗ | ✗ | 3.5K | ✓ | 1K | 10K |
| DetGPT (Pi et al., 2023) | DetGPT | SI | ✗ | ✓ | ✗ | 5K | ✗ | - | 30K |
| MULTIS (Zhao et al., 2023e) | ChatBridge | AA | ✓ | ✓ | ✗ | 80K | ✓ | 51K | 4.6M |
| pandagpt_vid (Su et al., 2023) | PandaGPT | AA+SI | ✓ | ✓ | ✗ | 83.5K | ✗ | - | 161.5K |
| PMC-VQA (Zhang et al., 2023f) | MedVInT-TE/TD | AA | ✓ | ✗ | ✗ | 149K | ✗ | - | 227K |
| GPT4Tools (Yang et al., 2023) | GPT4Tools | SI | ✓ | ✓ | ✗ | - | ✗ | - | 71K |
| M$^3$IT (Li et al., 2023k) | Ying-VLM | AA | ✓ | ✓ | ✗ | - | ✓ | - | 2.4M |
| MIMIC-IT (Li et al., 2023b) | Otter | SI | ✓ | ✓ | ✓ | 8.1M | ✓ | 502K | 2.8M |
| Video-ChatGPT (Maaz et al., 2023) | Video-ChatGPT | SI | ✓ | ✗ | ✗ | - | ✓ | - | 100K |
| MULTIINSTRUCT (Xu et al., 2022) | OFA$_{multiinstruct}$ | AA | ✓ | ✓ | ✗ | - | ✗ | - | 235K |
| Video-LLaMA (Zhang et al., 2023a) | Video-LLaMA | AA+SI | ✓ | ✗ | ✗ | 83.5K | ✓ | 11K | 164K |
| Valley-Instruct-73k (Luo et al., 2023b) | VALLEY | SI | ✗ | ✗ | ✗ | - | ✓ | 73K | 73K |
| MultiModal-GPT (Gong et al., 2023) | MultiModal-GPT | AA | ✓ | ✗ | ✗ | - | ✗ | - | 284.5K |
| InstructBLIP (Dai et al., 2023) | InstructBlip | AA | ✓ | ✓ | ✗ | - | ✓ | - | - |
| MACAW-LLM (Lyu et al., 2023) | MACAW-LLM | SI | ✓ | ✗ | ✗ | 10K | ✓ | 9.8K | 119K |
| LAMM (Yin et al., 2023) | LAMM | SI | ✓ | ✗ | ✗ | - | ✗ | - | 196K |
| OphGLM (Gao et al., 2023) | OphGLM | SI | ✓ | ✗ | ✗ | - | ✗ | - | 20K |
| LLaVAR (Zhang et al., 2023g) | LLaVAR | SI | ✓ | ✓ | ✗ | 16K | ✗ | - | 174K |
| Shikra-RD (Chen et al., 2023e) | Shikra | SI | ✗ | ✓ | ✗ | - | ✗ | - | 5.9K |
| PF-1M (Chen et al., 2023b) | Clever Flamingo | AA | ✓ | ✓ | ✗ | - | ✗ | - | 975K |
| GPT4RoI (Zhang et al., 2023e) | GPT4RoI | AA | ✗ | ✓ | ✗ | - | ✗ | - | - |
| SVIT (Zhao et al., 2023a) | SVIT(MMLLM) | SI | ✓ | ✓ | ✗ | 108.1K | ✗ | - | 3.2M |
| GRIT-20M (Peng et al., 2023) | KOSMOS-2 | AA | ✗ | ✓ | ✗ | - | ✗ | - | 20.5M |
| BuboGPT (Zhao et al., 2023c) | BuboGPT | AA+SI | ✓ | ✓ | ✗ | 83.5K | ✗ | - | 161.5K |
| MGVLID (Zhao et al., 2023c) | ChatSpot | AA+SI | ✓ | ✓ | ✗ | 1.2M | ✗ | - | 3M |
| Lynx (Zeng et al., 2023b) | Lynx | AA | ✓ | ✓ | ✗ | - | ✓ | - | - |
| Multimodal_id_v1 (Li et al., 2023m) | LMEye(IPN) | SI | ✓ | ✗ | ✗ | - | ✓ | - | 7.3M |
| AS-1B (Wang et al., 2023d) | ASM | AA+SI | ✗ | ✓ | ✗ | 11M | ✗ | - | 3.3B |
| I4 (Li et al., 2023g) | Cheetor | AA+SI | ✗ | ✗ | ✓ | 1.77M | ✗ | - | 477.7K |
| StableLLaVA (Li et al., 2023l) | StableLLaVA | AA | ✓ | ✓ | ✗ | - | ✗ | - | 126K |
| M-HalDetect (Gunjal et al., 2023) | - | SI | ✓ | ✗ | ✗ | 4K | ✗ | - | 16K |
| VIGC (Wang et al., 2023a) | VIGC | AA+SI | ✓ | ✓ | ✗ | - | ✗ | - | - |
| PointLLM (Xu et al., 2023b) | PointLLM | SI | ✓ | ✓ | ✗ | - | ✗ | - | 730K |
| CIEM (Hu et al., 2023a) | - | SI | ✓ | ✓ | ✗ | 4.9K | ✗ | - | 72.9K |
| GPTVQA (Zhao et al., 2023d) | MLLM-DataEngine | SI | ✓ | ✓ | ✗ | - | ✗ | - | 23K |
| T2M (Wu et al., 2023) | NExT-GPT | SI | ✗ | ✗ | ✗ | 4.9K | ✓ | 4.9K | 14.7K |
| MosIT (Wu et al., 2023) | NExT-GPT | SI | ✓ | ✗ | ✗ | 4K | ✓ | 4K | 5K |
| PVIT (Chen et al., 2023a) | PVIT | SI | ✗ | ✓ | ✗ | - | ✗ | - | 146K |
| TextBind (Li et al., 2023f) | TextBind | SI | ✓ | ✓ | ✓ | 63K | ✗ | - | 86.1K |
| DreamLLM (Dong et al., 2023) | DreamLLM | SI | ✓ | ✓ | ✓ | 120K | ✗ | - | 120K |
| AnyMAL (Moon et al., 2023) | AnyMAL | AA | ✓ | ✓ | ✓ | - | ✓ | - | 210K |
| InternLM-XComposer (Zhang et al., 2023b) | InternLM-XComposer | AA+SI | ✗ | ✗ | ✓ | - | ✗ | - | - |
| SparklesDialogue-VG (Chen et al., 2023a) | SparklesChat | SI | ✓ | ✓ | ✓ | 7K | ✗ | - | 4.4K |
| SparklesDialogue-CC (Chen et al., 2023a) | SparklesChat | SI | ✓ | ✓ | ✓ | 12.9K | ✗ | - | 2K |
| Ferret (You et al., 2023) | Ferret | AA+SI | ✗ | ✓ | ✗ | - | ✗ | - | 95K |
| MiniGPT-v2 (Chen et al., 2023d) | MiniGPT-v2 | AA+SI | ✓ | ✓ | ✗ | - | ✗ | - | - |
| ComVint (Du et al., 2023) | - | SI | ✓ | ✓ | ✗ | - | ✗ | - | 32K |
| GranD (Rasheed et al., 2023) | GLaMM | SI | ✓ | ✓ | ✗ | - | ✗ | - | - |
| LVIS-INSTRUCT4V (Wang et al., 2023b) | - | SI | ✓ | ✓ | ✗ | 110K | ✗ | - | 220K |

## A.3 Experimental Details

### A.3.1 Settings

In this paper, we used three different MLLMs to tune on VLIT data. Table 11, Table 12, and Table 13 respectively list all their hyperparameter settings during instruction tuning.

### A.3.2 Generation Details

**System Prompts.** Depending on the requirements for the generated command-response pairs, we use the following system prompts as setting input to GPT3.5.

You are an artificial intelligence assistant. You'll be given a text describing an image, and based on that text you'll be asked to generate a conversation around the object circled by a square box in the image. The conversation context is a question-and-answer session between the visual AI assistant and the user around the image. Specifically, the generated dialogue needs to meet the following requirements: 1. Answer confidently to the content described in the text. 2. The conversation must be about the object in the circle. 3. The visual artificial assistant's tone is that it sees the picture, not the text description. 4. It is necessary to ask more complex questions about the content in the image, appropriately combine it with external knowledge that does not appear in the image, and expand the discussion on the people, places, brands, and other information

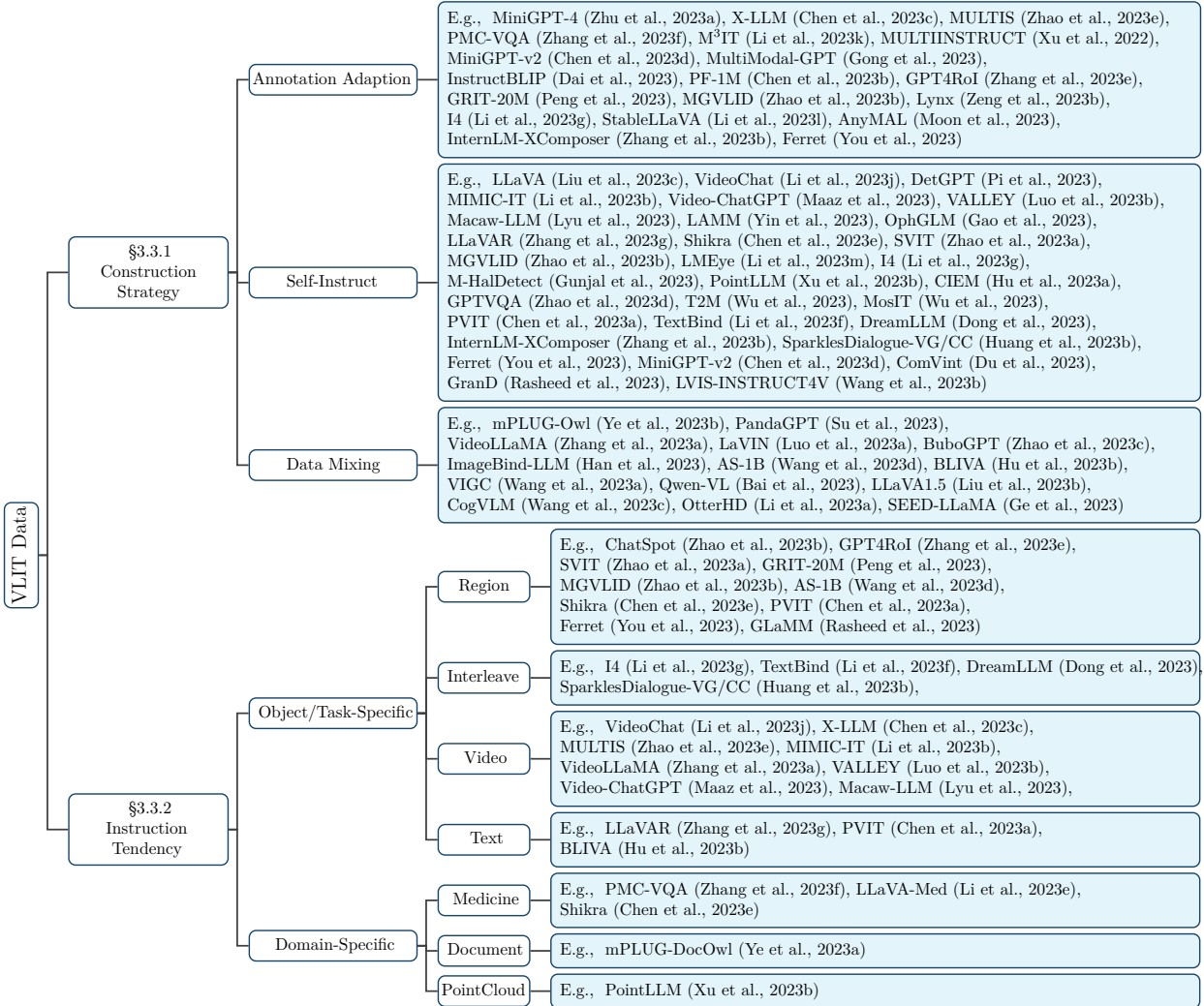

Figure 6: Tree category diagram of existing VLIT Data (The dataset without a name is replaced by its MLLM name).

known to the public in the image. 5. The answers in the conversation need to be as detailed as possible, and examples or reasoning steps need to be given when necessary to make the answer more convincing. 6. A certain number of negative answers need to be included in the generated dialogue, such as questions about non-occurring objects, and misdescriptions. 7. The generated conversation starts with the user asking a question. 8. Don't have anything like "the description", or "the given text" in the generated conversation. 9. The format of the output dialogue must refer to {"User": "Question?", "AI": "Answer."}.

**In-Context Examples.** In this paper, we generally obtain about 10 generated samples through manual writing, randomly select 2 of them as context, and input them to GPT3.5 in advance. The following is an example:

Here is an example, with the given textual description and the generated conversation. Given textual description: This is a picture which can be described as Two women preparing food in a kitchen, one at the sink and one at the table. A set table with silverware, glasses and a bottle of wine. A woman having a slice of bread with jelly. A woman sitting at a table with win and bread. A women who is tearing off a piece of bread. The image contains 2 kitchen_bottle, 1 furniture_dining table, 2 person_person, 5 kitchen_cup, 3 kitchen_fork, 4 kitchen_knife, 2 kitchen_spoon, 1 food_banana, 2 food_orange, 2 food_apple,

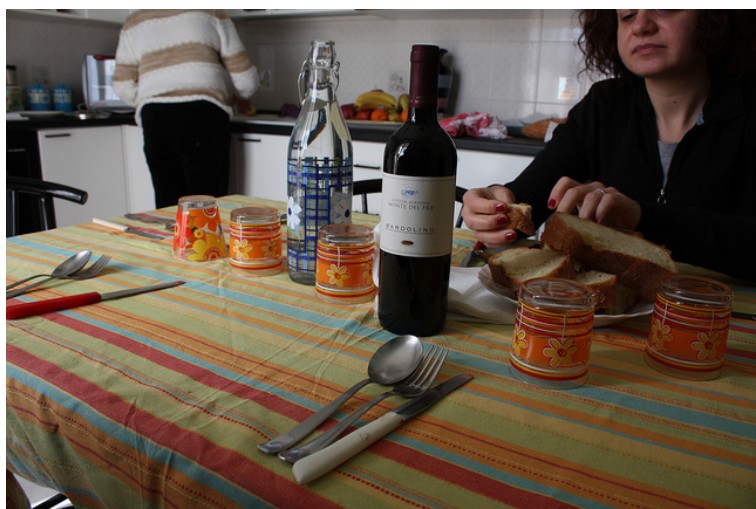

Figure 7: Image of in-context example.

2 furniture_chair. The image also contains a number of question and answer pairs: 'Q: IS there flowers on the table? A: no.', 'Q: How many bottles are on the table? A: 2.', 'Q: Is he woman making sandwiches? A: yes or no.', 'Q: How many glasses on the table are ready to receive a liquid immediately? A: 4.', 'Q: What type of patterns are featured? A: stripes.'. The following characters can be recognized in the image: BARDOLING. Generated conversation: {"User": "Are there any flowers on the table?", "AI": "No, there are no flowers on the table.", "User": "Is one of the women making sandwiches?", "AI": "One of the two women in the picture is actually making something, but it's not clear if it's a sandwich. The other person in the picture is handling bread.", "User": "Who do you think the women are in the picture?", "AI": "Based on the image alone, I cannot determine the identity of the women. However, it's possible that they could be friends or family members preparing a meal together.", "User": "What kind of fruit can you see in the picture?", "AI": "I see some bananas, oranges, and apples in the picture.", "User": "Can you tell me more about the furniture in the picture?", "AI": "The furniture in the picture includes a dining table and chairs."}

## B  Examples of Generated Instruction-Response Pair

Below, as shown in Table 14, Table 15, Table 16, Table 17, Table 18, and Table 19, we give several examples of instruction-response pairs generated in this paper.

## C  Correlation Analysis

The data in Table 20 is the detailed numerical content of the data in Figure 5.

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

Table 10: Visual task statistics in VLIT data.

| Name | Visual-Langage Task |
|------|---------------------|
| LLaVA (Liu et al., 2023c) | conversation, detailed description, complex reasoning |
| Mini-GPT4 (Liu et al., 2023c) | detailed description |
| mPLUG-Owl (Ye et al., 2023b) | conversation, detailed description, complex reasoning |
| VideoChat (Li et al., 2023j) | (video) conversation, (video) detailed description, complex reasoning |
| VisionLLM (Wang et al., 2023e) | object detection, instance segmentation, visual grounding, image captioning, visual question answering |
| X-LLM (Chen et al., 2023c) | (video) detailed description, visual question answering, dialogue generation |
| DetGPT (Pi et al., 2023) | image captioning, object detection |
| ChatBridge (Zhao et al., 2023e) | image captioning, visual question answering, (video) conversation, (video) detail description, (video) complex reasoning |
| PandaGPT (Su et al., 2023) | conversation, detailed description, complex reasoning |
| MedVInT-TE/TD (Zhang et al., 2023f) | visual question answering |
| GPT4Tools (Yang et al., 2023) | object grounding, object segmentation, image generating, visual question answering |
| LLaVA-Med (Li et al., 2023e) | visual question answering |
| Ying-VLM (Li et al., 2023k) | image captioning, visual question answering, reasoning, classification, image generation |
| Otter (Li et al., 2023b) | llava-interleaved, spot the difference, visual storytelling, dense captions, TV show captions, indoor event planning, ego4d |
| Video-ChatGPT (Maaz et al., 2023) | video reasoning, creation and generation, action recognition, video understanding, video conversation, visual question answering, temporal understanding |
| OFA$_{multiinstruct}$ (Xu et al., 2022) | visual question answering, visual relationship, temporal ordering, grounded generation, grounded matching, commonsense reasoning, miscellaneous, image understanding, region understanding, image text matching |
| Video-LLaMA (Zhang et al., 2023a) | (video) conversation, (video) detailed description, (video) complex reasoning |
| VALLEY (Luo et al., 2023b) | (video) conversation, (video) detailed description, complex reasoning |
| MultiModal-GPT (Gong et al., 2023) | conversation, detailed description, complex reasoning, image captioning, visual question answering |
| InstructBLIP (Dai et al., 2023) | image captioning, visual reasoning, conversation, visual question answering, image question generation, image classification |
| LaVIN (Luo et al., 2023a) | conversation, detailed description, complex reasoning, visual question answering |
| MACAW-LLM (Lyu et al., 2023) | image captioning, video captioning, visual question answering |
| LAMM (Yin et al., 2023) | conversation, complex reasoning, detailed description, visual task dialogue |
| OphGLM (Gao et al., 2023) | medical imaging description, causes and symptoms, diagnosis and examination, treatment and prevention, prognosis and lifestyle |
| LLaVAR (Zhang et al., 2023g) | visual question answering (text-rich) |
| Shikra (Chen et al., 2023e) | conversation, detailed description, complex reasoning, visual question answering, image captioning, grounding caption |
| Clever Flamingo (Chen et al., 2023b) | image captioning, image classification, change captioning, visual question answering |
| mPLUG-DocOwl (Ye et al., 2023a) | visual question answering, information extraction, natural language inference, image captioning, conversation, detailed description, complex reasoning |
| GPT4RoI (Ye et al., 2023a) | single-region captioning, multi-region captioning, single-region reasoning, multi-region reasoning, |
| SVIT (Zhao et al., 2023a) | conversation, detailed description, complex reasoning |
| KOSMOS-2 (Peng et al., 2023) | conversation, detailed description, complex reasoning |
| BuboGPT (Zhao et al., 2023c) | conversation, detailed description, complex reasoning |
| ChatSpot (Zhao et al., 2023b) | image captioning, visual question answering, object detection, OCR, region caption |
| ImageBind-LLM (Han et al., 2023) | conversation, detailed description, complex reasoning |
| Lynx (Zeng et al., 2023b) | visual question answering, image captioning, video captioning, classification, conversation, complex reasoning |
| LMEye(IPN) (Li et al., 2023m) | conversation, detailed description, complex reasoning, image-text matching |
| ASM (Wang et al., 2023d) | visual question answering, visual captioning |
| Cheetor (Li et al., 2023g) | conversation, visual relation inference, visual storytelling, multi-modal cloze, grounded question answering, text-rich question answering, complex reasoning |
| BLIVA (Hu et al., 2023b) | image captioning, visual question answering, conversation, OCR, detailed description, complex reasoning |
| StableLLAVA (Li et al., 2023l) | visual reasoning |
| Qwen-VL (Bai et al., 2023) | visual question answering, grounding, OCR |
| VIGC (Wang et al., 2023a) | conversation, detailed description, complex reasoning |
| PointLLM (Xu et al., 2023b) | brief description, conversation |
| MLLM-DataEngine (Zhao et al., 2023d) | visual question answering |
| MM (Tu et al., 2023) | conversation, detailed description, complex reasoning |
| NExT-GPT (Wu et al., 2023) | conversation |
| PVIT (Chen et al., 2023a) | small object recognition, same-category object discrimination, object relationship-based reasoning, object attribute-based reasoning, OCR |
| TextBind (Li et al., 2023f) | image creation, image comparison, intrinsic & extrinsic image understanding |
| DreamLLM (Dong et al., 2023) | conversation, detailed description, complex reasoning |
| AnyMAL (Moon et al., 2023) | visual question answering |
| InternLM-XComposer (Zhang et al., 2023b) | image captioning, visual question answering, image question generation, conversation, detailed description, complex reasoning |
| SparklesChat (Huang et al., 2023b) | conceptual caption, visual genome |
| LLaVA 1.5 (Liu et al., 2023b) | conversation, detailed description, complex reasoning |
| CogVLM (Wang et al., 2023c) | conversation, detailed description, complex reasoning, visual question answering |
| Ferret (You et al., 2023) | conversation, detailed description, complex reasoning |
| MiniGPT-v2 (Chen et al., 2023d) | visual question answering, captioning, grounded captioning, referring expression comprehension, referring expression generation, object parsing and grounding |

Table 11: Hyperparameters of LLaVA

| | |
|---|---|
| version | v1 |
| vision_tower | clip-vit-large-patch14 |
| pretrain_mm_mlp_adapter | LLaVA-7b-pretrain-projector-v1-1-LCS-558K-blip_caption.bin |
| mm_vision_select_layer | -2 |
| mm_use_im_start_end | False |
| mm_use_im_patch_token | False |
| fp16 | True |
| num_train_epochs | 1 |
| per_device_train_batch_size | 4 |
| per_device_eval_batch_size | 2 |
| gradient_accumulation_steps | 2 |
| learning_rate | 2e-5 |
| weight_decay | 0. |
| warmup_ratio | 0.03 |
| lr_scheduler_type | cosine |
| tf32 | False |
| model_max_length | 2048 |
| dataloader_num_workers | 4 |
| lazy_preprocess | True |

Table 12: Hyperparameters of BLIP-2

| | |
|---|---|
| model_type | caption_coco_opt2.7b |
| load_finetuned | False |
| use_grad_checkpoint | True |
| freeze_vit | False |
| lr_sched | linear_warmup_cosine_lr |
| init_lr | 1e-5 |
| min_lr | 0 |
| warmup_lr | 1e-8 |
| warmup_steps | 1000 |
| weight_decay | 0.05 |
| max_epoch | 1 |
| batch_size_train | 4 |
| num_workers | 4 |
| accum_grad_iters | 1 |
| max_len | 50 |
| min_len | 1 |
| num_beams | 3 |
| num_ans_candidates | 128 |
| inference_method | rank |
| seed | 42 |

Table 13: Hyperparameters of OpenFlamingo

| | |
|---|---|
| cross_attn_every_n_layers | 1 |
| dataset_resampled | True |
| batch_size | 1 |
| train_num_samples | 158000 |
| loss_multiplier | 0.2 |
| workers | 4 |
| freeze_lm_embeddings | True |
| run_name | OpenFlamingo-3B-vitl-mpt1b-us |
| num_epochs | 6 |
| warmup_steps | 1000 |

Table 14: An example of a conversation generated from an image as a whole.

| | |
|---|---|
| Image | 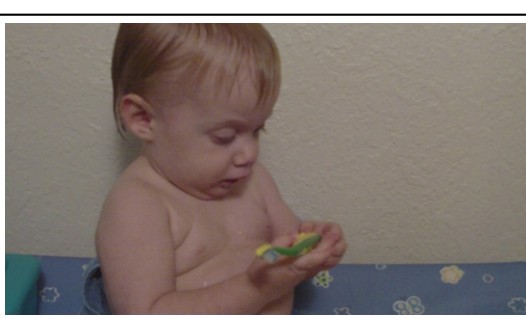 |
| User | Can you tell me more about the child in the picture? |
| AI Assistant | Certainly, the image showcases a small child, possibly a baby, who is engaging with a toothbrush. The child appears to be inspecting or playing with the toothbrush, which is colored green and yellow. The child seems to be feeling comfortable and cozy, and there's a sense of adorable curiosity about him. There's also a hint of seriousness in his actions, as if he's truly trying to figure out the purpose of the toothbrush. |
| User | How does the baby feel about the toothbrush? |
| AI Assistant | Based on the image, the baby seems intrigued by the toothbrush. The child doesn't show any signs of fear or discomfort; instead, he appears curious, as if he's trying to understand what the toothbrush is. It's a common behavior for children of this age to be curious about the objects around them. |
| User | Is the baby being fed in the photo? |
| AI Assistant | No, the image doesn't show any signs of the child being fed. The primary focus is on the child's interaction with the toothbrush. Any feeding utensils like spoons or bottles are not visible in the picture. |
| User | You mentioned the child looks serious. Can you elaborate on that? |
| AI Assistant | Sure, the term 'serious' here refers to the child's focused attention on the toothbrush. Even though he is just a baby, his intent gaze and the way he's holding the toothbrush suggest a sense of earnest exploration, as if he's trying to understand how this object works or what its purpose might be. It's an adorable demonstration of a child's natural curiosity and learning process. |
| User | Can you tell me about the toothbrush? |
| AI Assistant | Yes, the child is holding a toothbrush that is green and yellow in color. Given the size and the color combination, it seems like a typical toothbrush designed for children. It's quite common for children's toothbrushes to be colorful and attractive to help make the process of learning to brush their teeth more enjoyable. |
| User | Does the image suggest anything about the baby's mood? |
| AI Assistant | Yes, based on the attributes present in the image, the baby seems to be comfortable and curious. They look serious while inspecting the toothbrush, which suggests they are deeply engaged in the activity. Although there's no definitive way to tell, it's reasonable to say that the baby appears content and intrigued by their surroundings. |
| User | Is there anything else of note in the photo? |
| AI Assistant | The focus of this picture is mostly on the baby and their interaction with the toothbrush. There's no additional information about the surrounding environment. However, the fact that the baby feels comfortable playing naked suggests that they are probably in a familiar and safe space, most likely their home. |

Table 15: An example of a conversation generated from an image as a whole.

| | |
|---|---|
| Image | 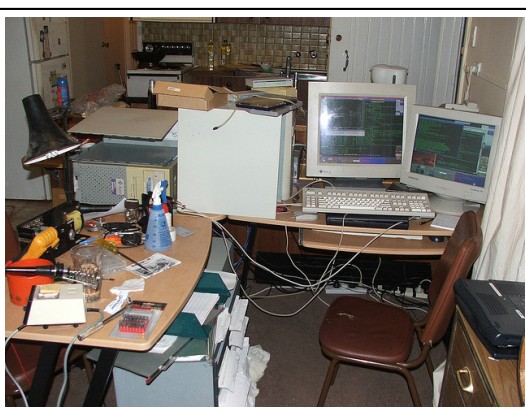 |
| User | Can you tell me more about the scene in this image? |
| AI Assistant | Absolutely! The image shows an office that is quite a bit cluttered. There is a desk equipped with two computer monitors, all of which are displaying various data. There's also additional computer equipment scattered around, including a laptop, a keyboard and couple of mice. There is a presence of other electronic devices as well, such as a TV. Interestingly, it's not just an office space, as I also see a sink and a refrigerator, and kitchenware like a cup, which suggests this might be a home office or a small startup office. The workspace seems to be quite disordered or messy. |
| User | Are the monitors up-to-date or old? |
| AI Assistant | The monitors in the image appear to be of an older model. |
| User | What can you tell me about the lighting for the workspace on the left? |
| AI Assistant | The workspace on the left seems to be using a lamp for illumination. It's quite common in home offices or personal workspaces to have such setup. |
| User | Is there any music-related equipment on the desk? |
| AI Assistant | From what I can observe, there might be a CD present on the tabletop, suggesting a possible interest in music or use of software stored on CDs by the person working here. |
| User | Would you describe this as an organized workspace? |
| AI Assistant | Yes, based on the attributes present in the image, the baby seems to be comfortable and curious. They look serious while inspecting the toothbrush, which suggests they are deeply engaged in the activity. Although there's no definitive way to tell, it's reasonable to say that the baby appears content and intrigued by their surroundings. |
| User | Is there anything else of note in the photo? |
| AI Assistant | Based on the scene in the image, this workspace does not appear to be organized. There are various pieces of equipment strewn about haphazardly, which suggests that the area is fairly messy. |
| User | Any idea what kind of work is being done in this office? |
| AI Assistant | Given the presence of soldering equipment and multiple computers, it could be an office where hardware development or repair is done. This might include tasks like assembling or fixing computer parts, or working with other electronic components. However, without more specific details in the image, it's difficult to say for certain. |
| User | Can you identify any other items in the picture? |
| AI Assistant | Sure, beyond the computer hardware, I can also see two chairs, indicating that there might be space for another person to work or meet. There is a keyboard and two mouses, suggesting that at least two computer systems are in use. There's a laptop present as well, which adds to the versatility of the workspace. Additionally, there's a lamp offering light, which can be particularly useful for detailed work like soldering. |

Table 16: An example of a generated conversation with negative responses.

| | |
|---|---|
| Image | 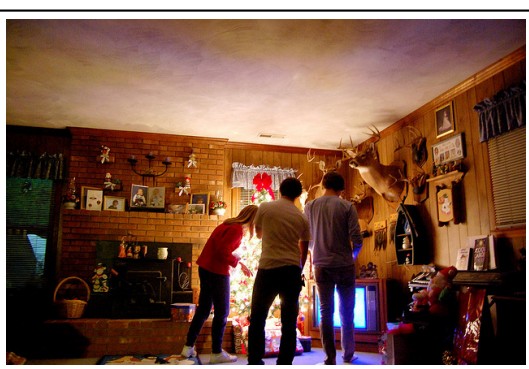 |
| User | Can you tell me what the people in the picture are doing? |
| AI Assistant | The people in the picture, who seem to be a family, are gathered around a television. They seem to be watching TV together in their living room. |
| User | What time of year does this scene seem to be set in? |
| AI Assistant | Given that there is a lit Christmas tree in the room, it strongly suggests that this scene is set during the holiday season, most probably around Christmas time. |
| User | Are there any books in the picture? |
| AI Assistant | Yes, there are indeed two books in the picture. |
| User | What's on top of the Christmas tree? |
| AI Assistant | The top of the Christmas tree is adorned with a bow. It's a common tradition to place a decorative item on the top of a Christmas tree, such as a star, angel, or in this case, a bow. |
| User | Are there any pets in the room? |
| AI Assistant | No, the description of the image doesn't mention any animals or pets being present in the room. |
| User | What are the kids looking at? |
| AI Assistant | The kids are looking out of a window. It's unclear what they're looking at specifically, as that detail isn't provided in the image. |
| User | Do you know what show they are watching on the TV? |
| AI Assistant | Unfortunately, the image doesn't provide specific information about what's being displayed on the television. They could be watching any number of things, such as a holiday-themed movie, a Christmas special of a television show, or even the news. |
| User | Is the Christmas tree the main focus of the image? |
| AI Assistant | While the Christmas tree is a significant element in the picture, it's not the main focus. The main focus appears to be the group of people, particularly the three kids looking out of a window and the family gathered around the television. |
| User | Are there any other significant objects in the picture? |
| AI Assistant | Yes, besides the Christmas tree and the television, the image also contains two books and a vase. However, it's not clear from the image how these objects relate to the people or the overall scene. |
| User | Are there any adults in the picture? |
| AI Assistant | From the information provided, it's not clear whether there are any adults in the picture. It mentions three people, who could be adults, children, or a combination of both. |

Table 17: An example of a generated conversation with negative responses.

| | |
|---|---|
| Image | 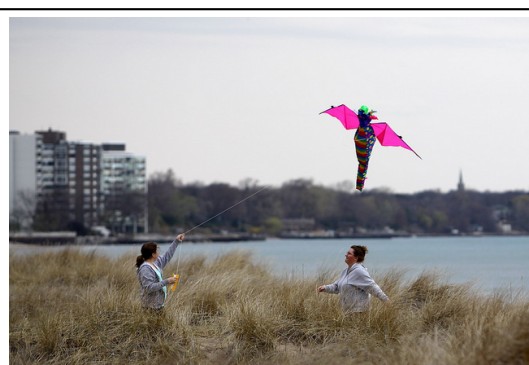 |
| User | What are the women doing in the picture? |
| AI Assistant | The women in the picture are enjoying their time in a field with tall grass where they are flying a kite that resembles a dragon. |
| User | Can you describe the kite in more detail? |
| AI Assistant | The kite in the picture has a design that makes it look like a dragon. Interestingly, its wings are reported to be pink. It sounds quite vibrant and playful. |
| User | Are the women in the image friends? |
| AI Assistant | Based on the image, both women appear to be friendly and are enjoying their time together. It's quite likely they are friends, though the image doesn't provide explicit confirmation of their relationship. |
| User | Does the field look like a public place? |
| AI Assistant | Yes, based on the image, it seems that the field is a public place. The women are dressed casually which also suggests that this is a relaxed, informal setting. |
| User | Do the women seem to be experts at flying kites? |
| AI Assistant | The image doesn't provide enough information to determine if they are experts at kite flying. However, since they seem to be enjoying themselves, it's clear they know at least the basics of how to fly a kite. |
| User | Is there any sign of a crowd or other people in the image? |
| AI Assistant | No, the available information doesn't mention the presence of other people or a crowd. The focus seems to be solely on the two women and their kite flying activity. |
| User | Do you know what season it is based on the image? |
| AI Assistant | The image doesn't give specific information about the season, though the presence of tall, dry grass might suggest it's late summer or early autumn. However, this is simply a conjecture based on the given image. |
| User | Are they in a park or a private property? |
| AI Assistant | The picture doesn't provide clear information about whether the field is part of a public park or private property. |
| User | What is the condition of the grass in the field? |
| AI Assistant | The grass in the field is quite tall and appears to be dry. It's probably in need of being cut, based on the information given. |
| User | Is there any wildlife in the picture? |
| AI Assistant | The image does not mention any wildlife. It primarily focuses on the two women and their activities with the kite in the field. |

Table 18: A generation conversation for regions where the image is marked.

| Image | 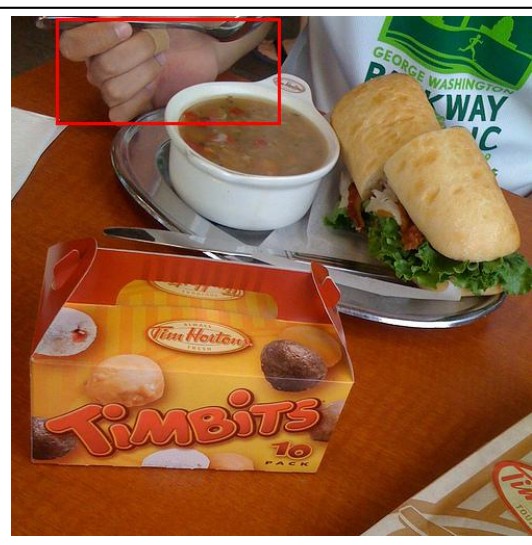 |
|---|---|
| User | What is the person in the circle doing? |
| AI Assistant | The person appears to be sitting and holding something, perhaps a utensil or one of the food items on the table. |
| User | Can we tell what the person is holding? |
| AI Assistant | From the image, it's uncertain exactly what the person is holding. It might be a spoon or knife. |
| User | Can you tell me something about the brand mentioned in the image? |
| AI Assistant | Yes, the brand Tim Hortons or Timbits is mentioned in the context of the image. Tim Hortons is a famous coffee shop chain in Canada known for its coffee and doughnut holes called 'Timbits'. It seems that the box of doughnut holes on the table might be from this brand. |
| User | Is the person alone or are there other people in the picture? |
| AI Assistant | From the given image, it appears that there's only one person visible in the image. However, without seeing the entire picture, it's hard to conclusively say if there are others present but not visible within the frame. |
| User | Do you think the person is about to eat the food? |
| AI Assistant | While it's difficult to definitively confirm without more context, the fact that the person is sitting at the table with a variety of food items in front of them, and appears to be holding something, suggests that they might be about to eat the food. However, they could also be in the process of setting the table or packing up leftovers. |
| User | Can we tell who this person is? |
| AI Assistant | As an AI, I can't identify individuals unless they are publicly recognizable and their information is part of my built-in knowledge. However, in this picture, the individual appears to be just a regular person sitting and holding something. |
| User | Are they eating healthy? |
| AI Assistant | The meal includes a sandwich and a soup, both of which have been described as healthy. However, they also have donut balls which are typically high in sugar and fats. So, while parts of their meal are healthy, the overall meal could be considered a mix of healthy and less healthy options. |
| User | Is there any beverage visible in the picture? |
| AI Assistant | No, there isn't any visible beverage in the picture. The image mostly showcases the food on the table, such as the sandwich, soup, and donuts, but doesn't show any drinks. |

Table 19: A generation conversation for regions where the image is marked.

| | |
|---|---|
| Image | 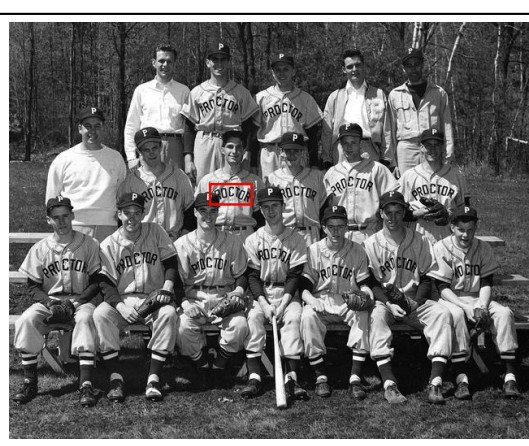 |
| User | What is marked in the image? |
| AI Assistant | The marked word in the image is 'PROCTOR'. |
| User | Could 'Proctor' denote the name of the baseball team? |
| AI Assistant | Yes, considering the background, 'Proctor' could indeed be the name of the baseball team depicted in the photograph. |
| User | Can you tell me more about the significance of 'Proctor' in relation to baseball? |
| AI Assistant | Unfortunately, I don't have any specific information about a baseball team named 'Proctor'. It could be a school team, a local community team, or even a professional team from the past. The name could also possibly refer to a location or sponsor. Without additional context, it's challenging to be certain. |
| User | Are there any well-known baseball players in the photo? |
| AI Assistant | Based on the given description, it is not possible to determine if there are any well-known players in this black-and-white photograph. The image features 14 individuals, presumably members of the 'Proctor' team, but without clear identification or further context, it's impossible to specify if any of them are famous. |
| User | Are all the people in the picture players? |
| AI Assistant | No, not everyone in the picture is a player. They may be coaches or team staff rather than players. |
| User | Is there any indication in the picture of the era or time period this game was played in? |
| AI Assistant | The fact that the photograph is in black and white suggests that it could be quite old, possibly from the early 20th century. However, without more context or specific details, it's hard to pinpoint an exact era. |
| User | Are there any spectators or fans visible in the picture? |
| AI Assistant | No, there are no spectators or fans visible in the picture. The image focuses on the baseball team named 'PROCTOR'. |

Table 20: Correlation coefficient data. The values in the table are correlation coefficients and p-values, respectively. Where ***, **, * represent significance levels of 1%, 5%, and 10%, respectively.

| | Overall | SU | II | IL | IA | IC | SR | IIR | VR | TR |
|---|---|---|---|---|---|---|---|---|---|---|
| MD | 0.952(0.048**) | 0.891(0.109) | 0.705(0.295) | 0.946(0.054*) | 0.852(0.148) | 0.929(0.071*) | 0.328(0.672) | 0.821(0.179) | 0.989(0.011**) | 0.999(0.001***) |
| TD | 0.932(0.068*) | 0.842(0.158) | 0.631(0.369) | 0.885(0.115) | 0.741(0.259) | 0.874(0.126) | 0.195(0.805) | 0.707(0.293) | 0.931(0.069*) | 0.972(0.028**) |
| ID | 0.986(0.014**) | 0.937(0.063*) | 0.778(0.222) | 0.965(0.035**) | 0.867(0.133) | 0.959(0.041**) | 0.402(0.598) | 0.728(0.272) | 0.971(0.029**) | 0.991(0.009***) |
| RD | 0.725(0.275) | 0.726(0.274) | 0.6(0.400) | 0.823(0.177) | 0.859(0.141) | 0.785(0.215) | 0.438(0.562) | 0.971(0.029**) | 0.893(0.107) | 0.827(0.173) |
| OC | 0.983(0.017**) | 0.959(0.041**) | 0.863(0.137) | 0.937(0.063*) | 0.838(0.162) | 0.951(0.049**) | 0.512(0.488) | 0.513(0.487) | 0.875(0.125) | 0.91(0.090*) |
| IC | 0.864(0.136) | 0.886(0.114) | 0.893(0.107) | 0.802(0.198) | 0.731(0.269) | 0.839(0.161) | 0.639(0.361) | 0.194(0.806) | 0.656(0.344) | 0.696(0.304) |
| Bal | 0.377(0.623) | 0.301(0.699) | 0.25(0.750) | 0.205(0.795) | 0.009(0.991) | 0.249(0.751) | -0.065(0.935) | -0.342(0.658) | 0.116(0.884) | 0.24(0.760) |

