# OpenReview forum: "Vision-Language Instruction Tuning: A Review and Analysis"
_TMLR — Accepted by TMLR_

### Review · Reviewer_hRhr · 2024-03-01

**Summary Of Contributions:**

This paper reviews the latest advances in vision-language instruction tuning (VLIT) to train multi-modal LLMs. It provides a detailed categorization for existing VLIT datasets and identifies several major design decisions necessary for training to succeed. The authors also supplement the review paper with experiments to verify their summarized insights on the performance of tuned multi-modal LLMs. Finally, they end with current challenges and future research directions.

**Audience:**

Yes

**Claims And Evidence:**

No

**Requested Changes:**

please see weaknesses above.

**Strengths And Weaknesses:**

strengths:
1. the paper is well motivated and certainly timely and relevant due to the recent this massive interest in large vision-language models.
2. the inclusion of experiments in a review paper is a big plus. there are careful ablations to test whether each of the 'recommended' strategies for VLIT actually translate to performance improvements, and these results can serve as useful insights for practitioners.
3. the paper is generally well-written and was nice to read, but some of the tables and figures can be improved to make them more readable and intuitive. see below.

weaknesses:
1. the writing and presentation of the paper can be improved. there should be an overview figure at the beginning highlighting the main taxonomy/categorization which is the main contribution of the paper. some of the tables and figures are too densely packed (such as table 1, table 2, and figure 2). these are quite hard for the reader to parse and is more suitable for the appendix. i am wondering if there can be simplified versions of table 1, table 2, and figure 2 that are more suitable for the main paper.
2. another aspect of presentation that can be improved is to have more visual depictions of the categories, instead of in purely words. for example, in table 2, the meaning of the categories 'annotation adaptation', 'self-instruct', 'data-mining' and so on are not immediately intuitive. is it possible to have some intuitive figures illustrating what they mean? and how they mainly differ from each other? also, why was this categorization chosen, and why is it the most suitable to categorize in this way? there are some attempts to draw these out in figure 3 and figure 4, but these figures are too dense and not clear at first glance. the figure captions should also be more detailed to assist the reader in parsing the figure. also, 'data-mining' is missing as a concept that is illustrated in a figure.
3. the coloring in figure 5 needs to be fixed. also, from the results in figure 5, does it mean that 'Balance' as a principle is not as important as the other concepts? even leading to negative correlation in some cases? also is there something about spatial relation that causes methods to not work? can the hypothesis 'This may be attributed to the fact that spatial positional relationships typically only have a certain correlation with the number of elements appearing in the instruction and the related instructions' be tested to see whether new guidelines are needed to improve in this case?
4. also, were the trials run several times and averaged/reported with error bars? how sensitive are some of these numbers to different runs?
5. is the only area of study in VILT in the type of data? are there are no other considerations such as in the training objectives or architectures of the models? if so, these should also be categories in the review paper and worth discussing. if not, please justify/argue why training objectives or architectures or other considerations are not included in the study.

---

> ### Author Response · Authors · 2024-03-30
> **Official Comment by Authors**
>
> **w1: the writing and presentation of the paper can be improved.**
>
> **response**: In the revised version, we first added an overview figure (Figure 1) that briefly presented the content of this survey. We also merged the original Table 1, Table 2, etc. into Table 1, and placed the corresponding detailed information in the appendix.
>
> **w2: another aspect of presentation that can be improved is to have more visual depictions of the categories, instead of in purely words.**
>
> **response**: In the revised version, we simplified Figure 3 for easier comparison and added examples and detailed captions for Figure 3 and Figure 4 (Please refer to the corresponding olive texts). Secondly, we also added a discussion on the classification principles (see olive texts in Sec. 3.2.1). In addition, we did not focus on discussing "Data Mixing" because it is a strategy that combines the first two construction schemes.
>
> **w3: Discussion of figure 5**:
>
> **response**: First, We have adjusted the discussion of Figure 5.
> As shown in Table 3, the "balance" evaluation metric is the lower the better (contrary to all other metrics), so it is reasonable that it is negatively correlated with performance in most cases. To avoid misunderstandings, we refined relevant descriptions in both the text and Table 3.
> I believe that there is a correlation between the performance of MLLM on "spatial relations" and the number of elements and related instructions mentioned in the input. This is because other instruction metrics (Y axis in Figure 5) usually do not involve discussions of spatial relationships (e.g., matching degree only considers overall semantic similarity, task diversity only considers the number of task types, etc.). The more objects that appear in the original data or are mentioned in the instructions, the more spatial relationships will be involved. Therefore, using it to train the model can improve the corresponding performance and serve as an accurate evaluation metric for assessing data performance on "spatial relations" (a correlation greater than 0.6 is considered a moderate or higher positive correlation). To verify this, I conducted a simple statistical analysis by randomly selecting 1,00 data samples with spatial relation instructions and 1,00 data samples without spatial relation instructions, and counted the number of objects in each sample. The results showed that spatial relation instructions really contained more objects.
>
> | instructions | number of objects |
> |-------|-------|
> | with spatial relation | 2.7 |
> | without spatial relation | 1.6 |
>
> **w4:  trials run several times**
>
> **response**:  Generally speaking, the performance of MLLMs does not require reporting errors, mainly because identical results can be obtained when the initial state of the model, training parameters, and dataset splitting are fixed. In addition, the VLIT data evaluation mentioned in this article, as most of the models used are CLIP, SIMCSE, Ego Splitting, etc. that do not contain random seeds, also do not have errors. For the selection of evaluation indicators such as LLM, as we mentioned in the article that multiple samples will be taken during LLM generation to ensure consistency of results, the error can be ignored for the average value, so it was not reported.
>
> **w5: is the only area of study in VILT in the type of data?**
>
> **response**: Thank you for your suggestion. We believe data is the most important part of VLIT. Additionally, model architecture and tuning settings are other factors that should be considered. Together, they can cover all study areas of VLIT. Therefore, we have added corresponding content to the revised version (see Table 1 and Sec. 3.1).
> Furthermore, the training objectives of the VLIT in MLLMs do not differ significantly between models and are discussed in the tuning settings and therefore do not need to be discussed separately.

---

### Review · Reviewer_ahPg · 2024-03-12

**Summary Of Contributions:**

In this paper, the authors provide an in-depth analysis of the design and characteristics of recent VLIT settings and datasets in multi-modal LLMs, providing insights into the motivations behind the dataset design. They give a detailed multi-perspective categorization of existing VLIT datasets.  Furthermore, they outline the qualities that high-standard VLIT data should possess.
This paper's significant contributions include:

-A systematic review of all relevant settings and datasets of VLIT in MLLMs. The authors describe the principles that are essential during the construction of VLIT data. They further highlight the ongoing challenges and future research areas that warrant further exploration.

-The paper proposes a detailed pipeline for creating high-quality VLIT data with explicit implementation details. This roadmap includes understanding the intrinsic motivations behind VLIT settings and pinpointing the qualities that high-standard VLIT data must have.

-The authors develop a VLIT dataset using the outlined construction pipeline and public annotation data. They compare this new dataset with prior VLIT datasets on varied MLLMs with distinct architectures. Their results substantiate the effectiveness of the principles and the usefulness of the construction pipeline.

-The authors make the code and dataset publicly available.

**Audience:**

Yes

**Claims And Evidence:**

Yes

**Requested Changes:**

-**Expansion on Multimodal Data Discussion, Specially for Vision data** (Critical): The paper would benefit from a more nuanced discussion on the distribution and mutual influence of multimodal versus unimodal data, the organization and composition of vision data, and the impact of high-quality vision data on visual instruction tuning.

-**Clarification on Pretraining Stage Description** (Critical): There is a need to clarify the pretraining stage description and discuss the relevant works that train the multi-model LLM from scratch.

-**Evaluation of CLIP Similarity Metric** (Strengthening): Given the limitations of clip similarity, the authors should provide further discussion and experimentation on the efficiency of using clip similarity as a metric for filtering instruction data.

-**Ablation Studies** (Critical): This would strengthen the paper's claims and provide clearer evidence of the benefits of their dataset construction pipeline.

-**Updating References** (Strengthening): Adding the missing discussions and references to some recent work could strengthen the impact and depth of this submission.

**Strengths And Weaknesses:**

[Strengths]

-The paper provides a comprehensive review of the latest developments in Vision-Language Instruction Tuning (VLIT) settings and datasets, offering valuable insights for both the community and researchers interested in the current state of multi-modal LLMs.

-It presents a detailed categorization of VLIT datasets from multiple perspectives, enhancing understanding of data requirements for effective multi-modal LLMs. From this insight, the authors propose a pipeline to create high-quality VLIT data that could improve visual instruction tuning for these models.

-The thorough experiments conducted demonstrate the effectiveness of the VLIT data produced by their proposed pipeline.

[Weaknesses]

-While the paper covers a wide range of VLIT settings and datasets, the paper doesn't significantly differ from the survey on language-only instruction tuning, which hinders the novelty of the paper. Given the modality gap between language and image, I think further discussion on topics such as the distribution and mutual influence of multimodal data and unimodal data, the organization and composition of vision data, the impact of high-quality vision data on visual instruction tuning, the influence of image data clarity and diversity, etc., would be beneficial.

-While much emphasis is placed on discussing the setting of Vision-Language Instruction Tuning, it appears that some aspects are still missing from the discussion. Specifically,  although the authors have divided the training process of Vision-Language Instruction Tuning into pretraining and finetuning stages, the pretraining stage described in the paper more closely resembles cross-modality alignment rather than pretraining. Given that there is extensive work focusing on pretraining Vision-Language models from scratch and treating Vision-Language Instruction Tuning as finetuning(e.g. Gemini[4], VideoPoet[5], Unified-IO1,2[7,8], Flamingo, etc). It seems that these works have been overlooked by the author.

-While the author discusses the correctness and diversity of instruction tuning data, the focus primarily remains on the language aspect. Further discussion of the fidelity and diversity of images would solidify this paper.

-The authors employ clip similarity as a metric to eliminate "incorrect" image-text pairs. Nonetheless, as suggested in [3], clip embedding similarity is already known to have limitations in capturing fine-grained, ambiguous information. Additionally, the instruction prompts generated by LLMs vary significantly from the training data of the clip model. Therefore, it necessitates further discussion and experimentation on the efficiency of filtering using clip similarity.

-Lack of ablation studies: Given the significant differences in the data organization and data sources between the LLava, MIMIC-IT datasets, and the generated VLIT dataset, it is challenging to draw a convincing conclusion about the superiority of the VLIT dataset without any ablation study.

-Some references are missing:

Page 5 Section 3.1 a  pixel-level linear projection [1]
Page 8 Section 3.2 A method of Multimodal in-context learning [2]
Page 12 Section 4 Correctness with clip  [3]

[1]  Rohan et al. fuyu-8b: Introducing our Multimodal Models

[2] Zhao et al. Mmicl: Empowering vision-language model with multi-modal in-context learning[J]. arXiv preprint arXiv:2309.07915, 2023.

[3] Tong et al. Eyes wide shut? exploring the visual shortcomings of multimodal llms[J]. arXiv preprint arXiv:2401.06209, 2024.

[4] Gemini: A Family of Highly Capable Multimodal Models

[5] Kondratyuk et al. Videopoet: A large language model for zero-shot video generation[J]. arXiv preprint arXiv:2312.14125, 2023.

[7] Lu et al. Unified-io: A unified model for vision, language, and multi-modal tasks[C]//The Eleventh International Conference on Learning Representations. 2022.

[8] Lu et al. Unified-io 2: Scaling autoregressive multimodal models with vision, language, audio, and action[J]. arXiv preprint arXiv:2312.17172, 2023.

---

> ### Author Response · Authors · 2024-03-30
> **Official Comment by Authors**
>
> **w1: Expansion on Multimodal Data Discussion, Specially for Vision data**
>
> **response**: Thank you for your suggestion. Unlike pre-training data (e.g., laion) from the Internet, the images in VLIT data often come from commonly used datasets (e.g., coco), ensuring the quality of the images. However, some deeper visual features may indeed affect the performance of VLIT. Therefore, in the revision, we added evaluations on image readability, diversity, and complexity, and conducted further discussions (please refer to red texts in Sec.4 and Sec. 5).
>
> **w2: Clarification on Pretraining Stage Description**
>
> **response**: Firstly, we have added some MLLMs trained from scratch in the revised version that were missing. Secondly, in this paper, we divided the fine-tuning process into two stages: pre-training (i.e., Feature Alignment) and VLIT, based on LLaVA and a large number of MLLMs that used the same training settings as LLaVA. We think the reason why most MLLMs refer to FA as "pre-training" is mainly because FA requires a massive amount of image-text pairs (such as LAION) for training and aims to guide the LLM to initially understand the semantic relationship between visual and textual information, similar to using ordinary text for pre-training in LLM. In addition, the vast majority of MLLMs trained from scratch also consider FA as pre-training, such as VideoPoet and Flamingo.
>
> **w3: Evaluation of CLIP Similarity Metric**
>
> **response**: We did notice this issue during the experiments (the generated instruction data had low and evenly distributed image-text similarity), which is widely present in the evaluation of tasks such as image editing, making it difficult to obtain an effective quantitative evaluation. Therefore, we have attempted to use more representation methods (such as DINO), and the results were consistent.
> Then, we manually observed 100 pairs of high- and low-similarity image-text instructions separately, and the results are shown in the following table. We found that there were many correct examples without problems in the low-similarity image-text instruction pairs, while there were almost no incorrect examples in the high-similarity image-text instruction pairs. Therefore, by weighing the cost and performance of filtering large amounts of data, the CLIP-based approach is a reasonable choice.
>
> |  | Positive | Negitive |
> |-------|-------|-------|
> | high-similarity | 98 | 2 |
> | low-similarity | 55 | 45 |
>
> **w4: Ablation Studies**
>
> **response**: Thank you for your suggestion. We conducted ablation experiments on the proposed filtering scheme, and the results are shown in Tables 2 and 3. By observing the results, it can be found that removing specific indicators leads to a significant decrease in the performance of some subtasks (such as diversity&instruction recognition), while removing the consideration of "balance" results in a significant differentiation in the performance of different tasks (tasks with a larger quantity improve performance, while tasks with a smaller quantity decrease performance). In addition, we have also made ablation settings for the impact of visual signals, as shown in Table 4, which reveals the most important quality requirement for readability of visual signals.
>
> **w5: Updating References**
>
> **response**: In the revised version, we have added the missing citations.

---

### Review · Reviewer_UyUM · 2024-03-20

**Summary Of Contributions:**

This paper aims at the literature review of multimodal large language models (MLLMs), specifically for vision-language instruction tuning (VLIT). They cover various VLIT settings as well as different constructions of VLIT datasets. Built upon this, they conduct experiments and demonstrate performance impacts from different VLIT settings.

**Audience:**

Yes

**Claims And Evidence:**

Yes

**Requested Changes:**

Please see the weakness for details.

+ Highlight different architectures of MLLM models.
+ An evaluation/comparison for each used VLIT dataset.
+ Concrete examples for the discussed limitations.

**Strengths And Weaknesses:**

**Strength**
+ This paper is well-organized and easy to follow.
+ The investigated VLIT is vital for building MLLMs, as previous studies mostly focus on the architecture.
+ The review is quite comprehensive (e.g., Tables 1&2), containing both models and datasets.

**Weakness**
+ We understand this paper focuses on VLIT. However, it is still required to highlight the model architecture (e.g., which LLM/vision-encoder/adapter is used) in Table 1, as they are all essential to make VLIT work and may influence performance. Besides, more discussion should be covered more clearly in the draft, instead of just a brief description (the first paragraph of Sec. 3.1).
+ Sec. 4 dives into the construction of VLIT datasets. As the proposed principles and quality aspects, why not have an experiment to evaluate all used datasets? This can be valuable to guide future research during dataset selection.
+ For the experiments, they involve LLaVA, BLIP-2, and OpenFlamingo. Though they attempt to use the same VLIT datasets, their inherent architectures are different (different LLM sizes or types of adapters). Furthermore, the pre-training data of their used LLMs are quite different (e.g., LLaMA in LLaVA is obviously better than OPT in BLIP-2). Is this comparison fair? Does the result actually reflect the quality of different VLIT datasets?
+ The discussed limitations (Sec. 6) are quite brief. It will be better to involve qualitative examples for each item, which can help them be more concrete and easy to understand.

---

> ### Author Response · Authors · 2024-03-30
> **Official Comment by Authors**
>
> **w1: Highlight different architectures of MLLM models.**
>
> **response**: Thank you for the suggestion. In the revised version, we incorporated a comprehensive discussion on the types of MLLMs' architectures. In addition, we also conducted a statistical analysis of the modules selected by the existing MLLM. Please refer to the blue texts in Sec. 3.1 and Table 1 of the revised version.
>
> **w2: An evaluation/comparison for each used VLIT dataset.**
>
> **response**: We have attempted the same settings as you mentioned during the initial experiment planning, but due to the slightly rough construction methods of the other two datasets we selected (as shown in Table 3) that did not fully consider the principles proposed in this paper, the data volume would be significantly reduced after filtering, making it difficult to support effective fine-tuning of MLLM. Therefore, the corresponding results were not reported. This is also one of the reasons why we proposed to build pipelines in this paper.
>
> However, in order to analyze the effectiveness of the proposed method in detail, we have added new ablation experiments, please refer to Tables 2, 3, and 4.
>
> **w3: Does the result actually reflect the quality of different VLIT datasets?**
>
> **response**: The reason why we chose multiple MLLMs is not to compare the performance between MLLMs, but to compare the performance of different quality VLIT data in different architectures and submodule environments. The experimental results are shown in Table 2, 3 and 4. The experimental results in each MLLM (llava, blip-2, and openflamingo) are consistent, indicating that the quality evaluation results of the VLIT dataset are directly proportional to the performance of MLLM after fine-tuning.
>
> **w4: Concrete examples for the discussed limitations.**
>
> **response**: Thank you for the suggestion. We added corresponding qualitative examples in the revised version. Please refer to the blue texts in Sec. 6.

---

### Author Response · Authors · 2024-03-30
**Official Comment by Authors**

We thank you for your time and effort in reviewing our paper.   We have responded to all comments in the rebuttal.   If you have any other comments or questions, please let us know.

Thank you for your attention.

---

### Decision · Action_Editor_tw6e · 2024-06-25

**Recommendation:** Accept as is

**Comment:**

The paper provides a comprehensive survey of visual language instruction tuning. The paper is well organized, including both methods and datasets.

**Audience:**

Researchers working on visual language models.

**Claims And Evidence:**

The paper provides a comprehensive survey of visual language instruction tuning.

---

> ### Author Response · Authors · 2024-07-13
> **Thanks!**
>
> We thanks the reviewers and action editors for the helpful feedback! We've uploaded a camera ready version that has taken into account all feedback.
>
> Best, Authors